# Impairing one sensory modality enhances another by reconfiguring peptidergic signalling in *Caenorhabditis elegans*

**Giulio Valperga[1,2], Mario de Bono[1,2]***

[1]Cell Biology Division, MRC Laboratory of Molecular Biology, Cambridge, United Kingdom; [2]Institute of Science and Technology Austria (IST Austria), Klosterneuburg, Austria

**Abstract** Animals that lose one sensory modality often show augmented responses to other sensory inputs. The mechanisms underpinning this cross-modal plasticity are poorly understood. We probe such mechanisms by performing a forward genetic screen for mutants with enhanced $O_2$ perception in *Caenorhabditis elegans*. Multiple mutants exhibiting increased $O_2$ responsiveness concomitantly show defects in other sensory responses. One mutant, *qui-1*, defective in a conserved NACHT/WD40 protein, abolishes pheromone-evoked $Ca^{2+}$ responses in the ADL pheromone-sensing neurons. At the same time, ADL responsiveness to pre-synaptic input from $O_2$-sensing neurons is heightened in *qui-1*, and other sensory defective mutants, resulting in enhanced neuro-secretion although not increased $Ca^{2+}$ responses. Expressing *qui-1* selectively in ADL rescues both the *qui-1* ADL neurosecretory phenotype and enhanced escape from 21% $O_2$. Profiling ADL neurons in *qui-1* mutants highlights extensive changes in gene expression, notably of many neuropeptide receptors. We show that elevated ADL expression of the conserved neuropeptide receptor NPR-22 is necessary for enhanced ADL neurosecretion in *qui-1* mutants, and is sufficient to confer increased ADL neurosecretion in control animals. Sensory loss can thus confer cross-modal plasticity by changing the peptidergic connectome.

**\*For correspondence:** mario.debono@ist.ac.at

**Competing interest:** The authors declare that no competing interests exist.

## Editor's evaluation

In this study Valperga and de Bono make the intriguing observation that interfering with the sensory function of a nociceptive neuron, termed ADL, alters its gene expression programs causing a recon-figuration of its functions. Upon loss of its properties as a primary sensor ADL gets repurposed by oxygen sensory circuits to enhance neurosecretion in an environmental oxygen dependent manner; thereby it adopts some interneuron like properties. The study is an interesting example of cross modal plasticity in neuronal circuits. It enables future studies on the ethological function of this phenomenon.

## Introduction

Animals that lose a sensory modality often show increased sensitivity to other sensory inputs. This change can involve repurposing neurons or brain areas that normally mediate responses to the lost modality such that they process other sensory inputs. For example, in blind people the absence of visual stimulation leads to rewiring of inputs into primary visual cortex (V1), so that V1 becomes respon-sive to tactile stimuli, a characteristic absent in sighted individuals (*Büchel et al., 1998*; *Dietrich et al., 2013*; *Sadato et al., 1996*; *Wanet-Defalque et al., 1988*). The molecular mechanisms enabling such

repurposing of neural circuits are incompletely understood, but at some level are thought to reflect opportunities for rewiring.

Animals can execute innate behaviours without a need for prior learning. However, experience and context can modulate innate behaviours, with circuits coordinating innate responses integrating information from modulating sensory pathways. Connections that link circuits mediating responses to distinct sensory cues provide opportunities to re-route sensory information if one sensory pathway is damaged (*Fine and Park, 2018*). *Caenorhabditis elegans* provides a favourable model to study cross-modal interactions in neural circuits, and how these connections may be altered by neural plasticity, in particular because of the careful reconstruction of its complete wiring diagram of chemical and electrical synapses (*Cook et al., 2019*; *Jarrell et al., 2012*; *White et al., 1986*). These studies have emphasized the anatomical stereotypy of the *C. elegans* nervous system, which contrasts with extensive experience-dependent plasticity at the behavioural level (*Fenk and de Bono, 2017*; *Pocock and Hobert, 2010*; *Saeki et al., 2001*; *Zhang et al., 2005*).

A salient environmental cue for *C. elegans* is oxygen ($O_2$) levels (*Gray et al., 2004*; *Persson et al., 2009*; *Zimmer et al., 2009*). Instantaneous as well as prior $O_2$ experience can reconfigure the value of sensory cues for this animal. For example, animals acclimated to 21% $O_2$ are attracted to pheromones that repel animals acclimated to 7% $O_2$ (*Fenk and de Bono, 2017*). The wiring diagram, coupled with $Ca^{2+}$ imaging, provides tantalizing hints about the basis of cross-modal plasticity associated with changes in $O_2$ levels. One of the main $O_2$-sensing neurons, URX, forms a spoke in a large hub-and-spoke circuit centred on the RMG interneurons (*Macosko et al., 2009*). Several sensory neurons, including pheromone receptors called ASK and ADL that, respectively, mediate attraction and repulsion from pheromones, form additional spokes in the circuit (*Jang et al., 2012*; *Macosko et al., 2009*). The URX $O_2$ sensors show persistent higher activity at 21% $O_2$ compared to 7% $O_2$, and tonically transmit this activity to the RMG hub interneurons (*Busch et al., 2012*). These $O_2$-evoked changes in URX and RMG somehow alter the pheromone response properties of ASK and ADL (*Fenk and de Bono, 2017*). Reciprocally, altering sensory transduction in the ASK or ADL neurons influences how *C. elegans* responds to $O_2$ stimuli (*de Bono et al., 2002*; *Laurent et al., 2015*; *Macosko et al., 2009*). However, the molecular underpinnings of how cross-modal changes are coordinated across the hub-and-spoke circuit as different elements of the circuit become more or less active are unclear.

Here, we employ forward genetics to identify mechanisms that alter information processing across the RMG hub-and-spoke circuit. We suppress *C. elegans* arousal in response to 21% $O_2$ by using genetic backgrounds that reduce signalling from RMG. We then seek mutants that restore $O_2$ responsiveness; such mutants are likely to reprogram sensory information processing across the hub-and-spoke circuit to circumvent RMG inhibition. We identify several sensory defective mutants that increase ADL's ability to relay information from pre-synaptic neurons, including from URX $O_2$ sensors and RMG interneurons. Specifically, these mutants show increased $O_2$-evoked secretion of neuropeptides from ADL. Using RNA sequencing (RNAseq), we profile ADL neurons in wild-type control and one enhancer mutant, *qui-1*. We discover extensive remodelling of ADL's peptidergic properties and find that increased expression of the neuropeptide receptor NPR-22 is necessary and sufficient to increase neurosecretion from ADL. Our data suggest that defects in sensory perception by the ADL pheromone sensors can increase ADL's responsiveness to input from the $O_2$ circuit by reconfiguring its sensitivity to neuropeptides. Changes in the peptidergic connectome may be an unappreciated mechanism by which loss of one sensory modality alters responsiveness to another.

## Results

### A genetic screen for enhancers of *C. elegans* aggregation behaviour

Natural isolates of *C. elegans* avoid and escape 21% $O_2$ (*de Bono and Bargmann, 1998*; *Cheung et al., 2004*; *Gray et al., 2004*; *Persson et al., 2009*). On a bacterial lawn these animals move rapidly and continuously while seeking lower $O_2$ concentrations such as areas of thick bacterial growth. A hub-and-spoke network that integrates multiple sensory cues coordinates this escape behaviour (*Figure 1A* and *Fenk and de Bono, 2017*; *Laurent et al., 2015*; *Macosko et al., 2009*). The standard *C. elegans* lab strain N2 (Bristol), referred to here as wild-type (WT), is not aroused by 21% $O_2$ and does not accumulate on thick bacteria. This is due to a gain-of-function mutation in the neuropeptide receptor NPR-1, *npr-1* 215V (*de Bono and Bargmann, 1998*) which arose during domestication of the

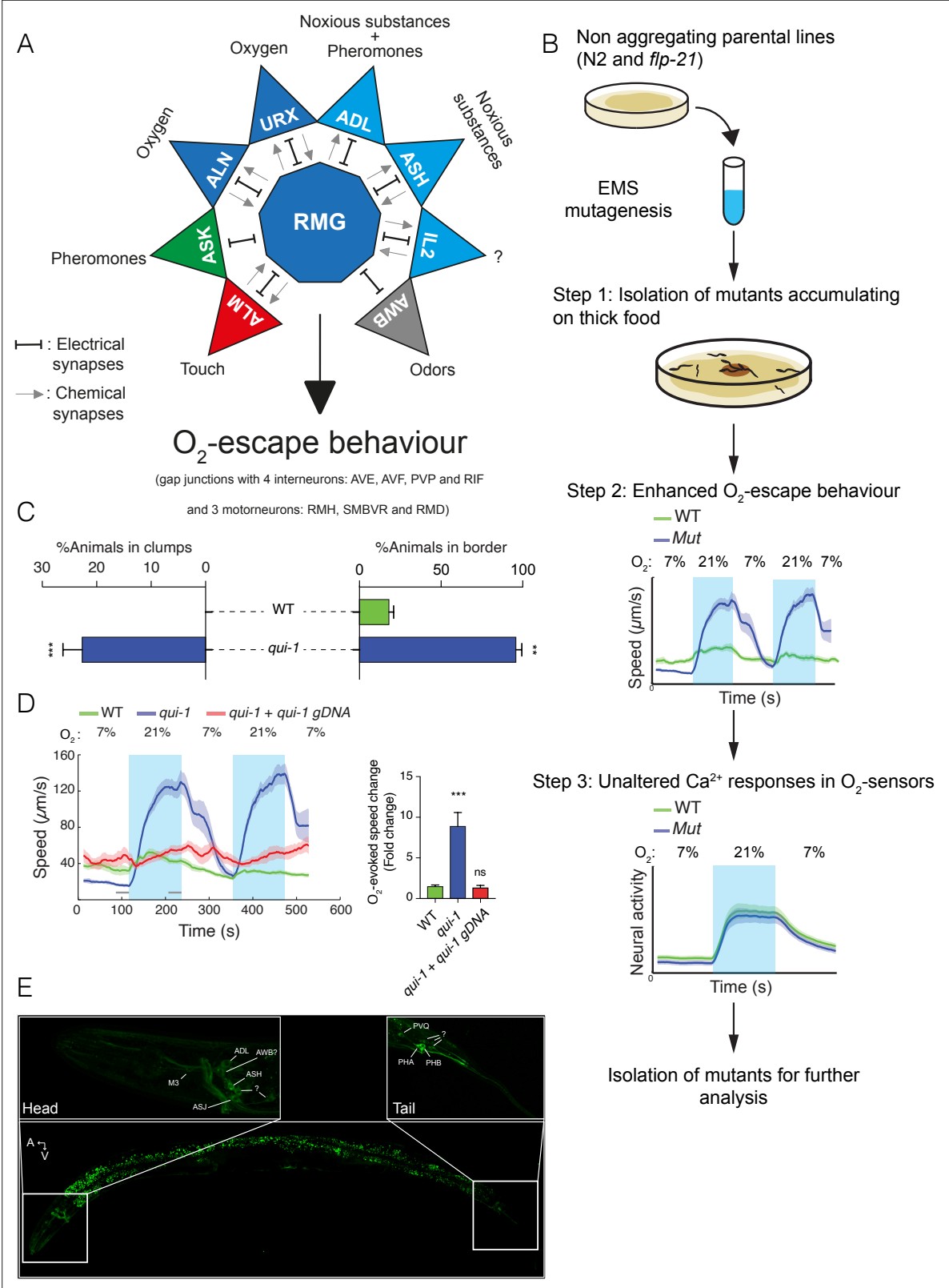

**Figure 1.** A genetic screen for mutants with enhanced $O_2$-escape behaviour. (**A**) The hub-and-spoke circuit associated with the URX $O_2$ sensors and $O_2$-escape behaviour, updated according to **Cook et al., 2019**. (**B**) Schematic of the genetic screen. We selected mutants that preferentially accumulate on thicker bacteria, a behaviour that depends on $O_2$ responses (Step 1), screened these mutants for increased $O_2$-escape behaviour (Step 2), and then identified strains with overtly normal $O_2$-evoked $Ca^{2+}$ responses in the URX $O_2$ sensors and RMG interneurons (Step 3). (**C**) Bar graphs quantifying

*Figure 1 continued*

aggregation and bordering behaviour. $N$ = 4–6 assays. (**D**) A wild-type copy of *qui-1* rescues the $O_2$-escape phenotype of *qui-1(db104)* mutants. Left: line shows average speed, shading shows standard error of the mean (SEM), and grey bars show the 30s time intervals used to calculate the average speed at 7% and 21% $O_2$. Right: the bar graph shows the fold change in average speed at 21% $O_2$ compared to 7% $O_2$. $N$ = 6–9 assays. (**E**) QUI-1 expression and localization using an *mNeonGreen::qui-1* translational fusion knock-in strain. Fluorescent neurons include ADL, ASH (Head) and PVQ, PHB and PHA (Tail), and potentially M3, AWB, and ASJ based on position and morphology. Also visible is yellow gut autofluorescence. Statistics: **p value ≤0.01; ***p value ≤0.001; ns, not significant, Mann–Whitney *U*-test. Comparisons are with wild-type. (**C, D**) Error bars represent standard error of the mean.

The online version of this article includes the following figure supplement(s) for figure 1:

**Figure supplement 1.** Characterization of *qui-1* mutants.

original Bristol wild strain (*McGrath et al., 2009*; *Weber et al., 2010*). NPR-1 215V inhibits $O_2$-escape behaviour by acting in the RMG interneurons to reduce the output of the hub-and-spoke circuit (*Macosko et al., 2009*). We exploited this inhibition to set up genetic screens seeking mutants that restored $O_2$-escape behaviour to *npr-1 215*V animals. Such mutants are likely to circumvent RMG's inhibition by re-routing sensory information and increasing the output of the hub-and-spoke circuit, thereby shedding light on cross-modulation of sensory circuits.

The NPR-1 receptor has multiple peptide ligands, including FLP-21 (*Rogers et al., 2003*). Deleting *flp-21* is not sufficient to restore $O_2$-evoked behaviours to N2 animals, but enhances $O_2$ escape in some contexts (*Laurent et al., 2015*; *Rogers et al., 2003*). We therefore mutagenized both N2 animals, and *flp-21* deletion mutants, and selected for individuals that accumulated preferentially on a patch of thick bacteria (OP50) placed in the middle of a thin lawn (see Methods) (*Figure 1B* – Step 1). We isolated 22 mutants from the N2 parental strain and 17 mutants from the *flp-21* parental strain that preferentially accumulated on the thick food patch. From these mutants, we further selected six strains that displayed enhanced $O_2$-evoked changes in locomotory activity compared to N2 controls (*Figure 1B* – Step 2). N2 and *flp-21* animals show only a modest change in locomotory activity when $O_2$ levels change from 7% to 21%, due to reduced RMG activity. By contrast, animals with a functional $O_2$ circuit become aroused at 21% $O_2$ and quiescent at 7% $O_2$ (*Busch et al., 2012*). To capture these differences in locomotory activity in one metric we plotted the ratio between animal speed at 21% and 7% $O_2$.

To identify the genetic defects causing increased $O_2$-escape behaviour in these mutants we used a Deep Sequence Mapping strategy (*Zuryn et al., 2010*). A list of de novo high impact mutations highlighted a premature stop codon (Q966Stop) within the *qui-1* gene in a mutant from the N2 parental strain (*Figure 1—figure supplement 1A*). Previous work suggested *qui-1* mutants lay eggs where bacteria are thickest (*Neal et al., 2016*). *qui-1(db104)* mutants isolated in our screen displayed both aggregation and $O_2$-escape behaviour (*Figure 1C, D*). We next compared the $O_2$-escape behaviour of the *db104* mutant with a strain carrying a deletion allele, *qui-1(ok3571)* (*Figure 1—figure supplement 1A*). These strains showed indistinguishable responses (*Figure 1—figure supplement 1B*), further suggesting that disrupting *qui-1* confers strong $O_2$-escape behaviour. To confirm this, we showed that a wild-type *qui-1* transgene completely rescued the *qui-1(db104)* $O_2$-escape phenotype (*Figure 1D*).

In the hub-and-spoke circuit, the URX $O_2$ sensors are tonically activated by 21% $O_2$ and in turn tonically activate the RMG hub interneurons (*Busch et al., 2012*). Optogenetic experiments show that increasing URX or RMG activity is sufficient to stimulate rapid movement (*Busch et al., 2012*). Two additional $O_2$ sensors, AQR and PQR, while not a part of the hub-and-spoke circuit, also signal increasing $O_2$ concentrations to the animal. To probe the *qui-1* phenotype, we imaged $O_2$-evoked $Ca^{2+}$ responses in the URX, AQR, PQR, and RMG neurons in *qui-1* mutants (*Figure 1B* – Step 3). *qui-1* $Ca^{2+}$ responses in each of these neurons resembled those of wild-type controls (*Figure 1—figure supplement 1C–F*), suggesting that the augmented $O_2$-escape behaviour of *qui-1* animals does not reflect a simple increase in the activity of $O_2$ sensors or RMG interneurons.

## The NACHT/WD40 protein QUI-1 acts in the ASH and ADL spoke neurons to inhibit $O_2$-escape behaviour

Previous work (*Hilliard et al., 2004*) and homology searches suggest QUI-1 is an ortholog of NWD1 (Nacht and WD40 repeat domain containing 1), a conserved protein of poorly understood function (*Figure 2—figure supplement 1A, B*). The *C. elegans* genome also encodes a paralog of QUI-1, T05C3.2, most similar to mammalian NWD2 (*Figure 2—figure supplement 1B*). These proteins

combine a NACHT domain with multiple WD40 domains and have homologs across phylogeny (**Figure 2—figure supplement 1B, C**). WD40 domains mediate protein–protein or protein–DNA interactions. NACHT domains are present in proteins involved in programmed cell death and transcription of the major histocompatibility complex, and include an NTPase domain, which is proposed to regulate signalling from these proteins. Most of the Walker A motif (Motif 1P loop) in the NTPase domain, which binds nucleotides, is conserved in QUI-1 (**Figure 2—figure supplement 1C**), suggesting the NACHT domain is functional.

Previous work suggests *qui-1* is expressed in a small subset of sensory and interneurons (**Hilliard et al., 2004**). To confirm the *qui-1* expression pattern, we used CRISPR/Cas9 genome editing to insert DNA encoding the mNeonGreen fluorescent protein in frame with the N terminus of QUI-1. Fluorescence from the mNeonGreen::QUI-1 fusion protein was confined to head and tail neurons, and we observed expression in ASH, ADL, PHB, and PVQ as previously reported (**Figure 1E** and **Hilliard et al., 2004**). We observed expression in five additional neurons close to the nerve ring, including possibly M3, AWB, and ASJ, and three neurons in the tail (**Figure 1E**). The mNeonGreen::QUI-1 fusion protein appears to be largely cytosolic and excluded from the nucleus, consistent with previous reports (**Figure 1E** and **Neal et al., 2016**).

Two of the *qui-1*-expressing neurons, ASH and ADL, form part of the RMG hub-and-spoke circuit (**Macosko et al., 2009**). ASH and ADL have previously been shown to promote aggregation and escape from 21% $O_2$ (**de Bono et al., 2002**), although they are probably not primary $O_2$ sensors. ASH and ADL are nociceptors that mediate *C. elegans* avoidance from a variety of chemical and non-chemical stimuli (**Hilliard et al., 2005**; **Jang et al., 2012**), for example $Cu^{2+}$ (ASH/ADL) and pheromones (ADL). We used cell-specific rescue of *qui-1* mutants to ask if QUI-1 acts in ASH and/or ADL neurons to inhibit $O_2$-evoked escape behaviour. Expressing *qui-1* selectively in ASH neurons reduced the $O_2$-escape response of *qui-1* mutants compared to wild-type animals (**Figure 2A**). The rescue was not complete: transgenic animals retained a significant $O_2$ response compared to wild-types (**Figure 2A**). Expressing *qui-1* only in ADL also significantly reduced the $O_2$-evoked escape behaviour of *qui-1* mutants (**Figure 2B**), but as with targeted expression in ASH, rescue was incomplete and transgenic animals responded significantly more to a 21% $O_2$ stimulus than wild-type animals (**Figure 2B**). Expressing QUI-1 in both ASH and ADL neurons did not show an additive rescue effect (**Figure 2C**), consistent with ablation studies suggesting these neurons act redundantly to promote aggregation behaviour (**de Bono et al., 2002**). We conclude that QUI-1 acts in ASH, ADL, and potentially other neurons to downregulate $O_2$-escape behaviour.

## QUI-1 is required for pheromone-evoked $Ca^{2+}$ responses in ADL

*qui-1* mutants exhibit chemosensory response defects (**Hilliard et al., 2004**; **Neal et al., 2016**), but QUI-1's role in these responses is not understood. Since $O_2$ signalling remodels the hub-and-spoke circuit, including ADL neurons (**Fenk and de Bono, 2017**), we speculated that disrupting *qui-1* alters ADL properties in a way that enhances circuit output in response to $O_2$ stimuli. To probe how loss of *qui-1* alters ADL function, we first examined ADL responses to pheromones. In wild-type control animals ADL neurons responded to the C9 ascaroside pheromone with a $Ca^{2+}$ response, as expected (**Jang et al., 2012**), however this response was completely abolished in *qui-1* mutants (**Figure 2D**). This suggests that QUI-1 is required for sensory transduction of pheromone stimuli.

ADL neurons promote escape from 21% $O_2$ (**de Bono et al., 2002**; **Laurent et al., 2015**). Consistent with this, *npr-1* mutants display a rise in $Ca^{2+}$ in ADL neurons in response to a 21% $O_2$ stimulus (**Fenk and de Bono, 2017**). This ADL $Ca^{2+}$ response depends on the URX neurons and the GCY-35/GCY-36 soluble guanylyl cyclases that are the primary $O_2$ sensors in these neurons, and is not detectable in N2 animals (**Fenk and de Bono, 2017**; **Zimmer et al., 2009**). To investigate if disrupting *qui-1* altered $O_2$-evoked $Ca^{2+}$ responses in ADL, we imaged these responses using GCaMP6s, which provides improved sensitivity compared to GCaMP3 (**Chen et al., 2013**). GCaMP6s reported a small but robust rise in $Ca^{2+}$ upon stimulation with 21% $O_2$ in N2 control animals, which rapidly returned to baseline (**Figure 2E**). Our ability to detect an $O_2$-evoked response in ADL in N2 likely reflects the improved sensitivity of GCaMP6s compared to the GCaMP3 used previously (**Fenk and de Bono, 2017**). Surprisingly, loss of *qui-1* abolished ADL $O_2$-evoked $Ca^{2+}$ responses, and stimulation with 21% $O_2$ resulted, if anything, in a reduction of ADL's $Ca^{2+}$ levels (**Figure 2E**). These data suggest that a

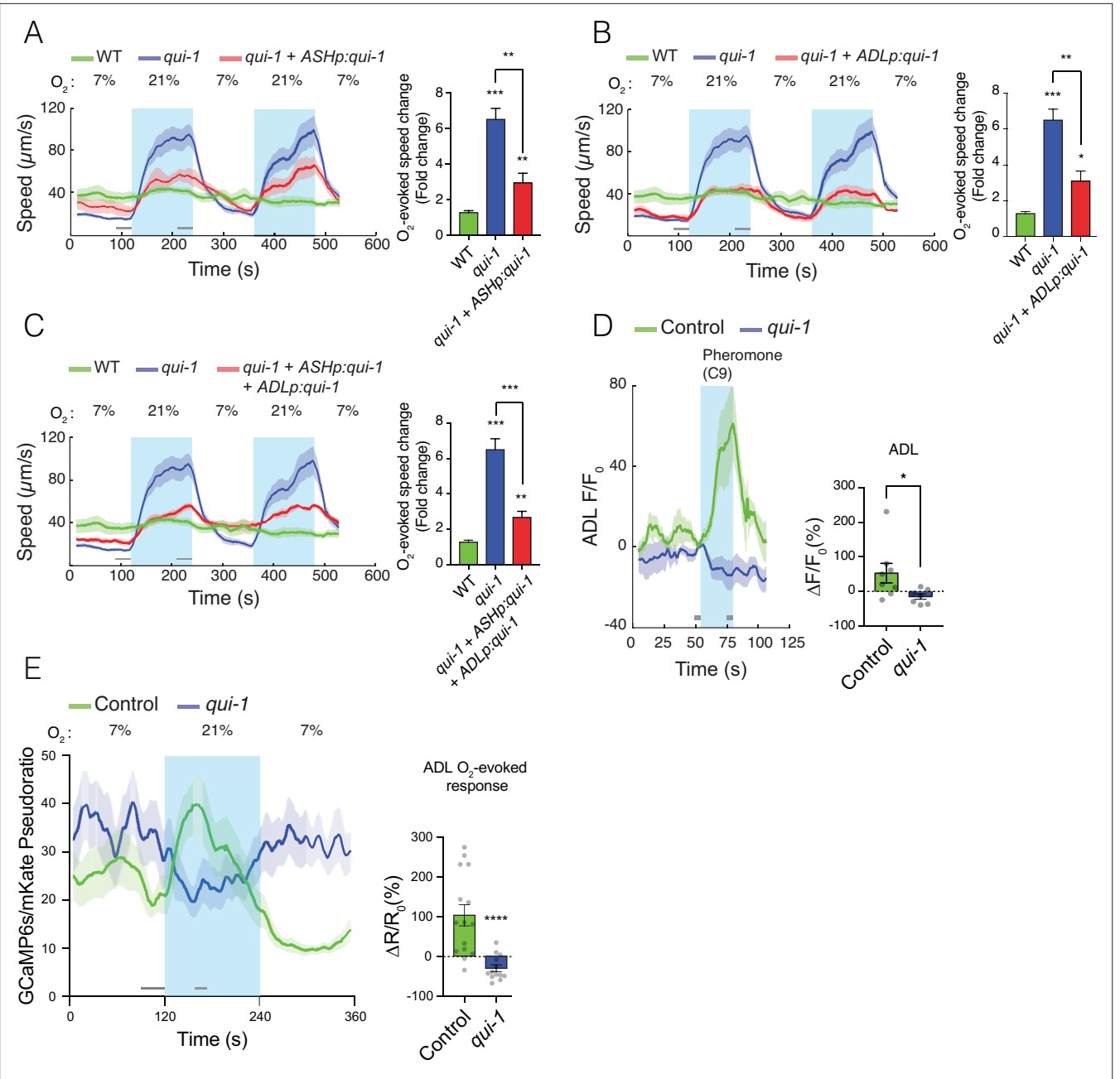

**Figure 2.** *qui-1* acts in ADL and ASH chemosensory neurons to inhibit $O_2$-escape behaviour, and is required for pheromone-evoked $Ca^{2+}$ responses in ADL. (**A–C**) Selective expression of *qui-1* in ASH (*sra-6p*), ADL (Δ*sre-1p*), or ASH+ADL (*sra-6p* + Δ*sre-1p*) neurons partially rescues the $O_2$-escape phenotype of *qui-1* mutants. Left: lines show average speed, while shading represents standard error of the mean (SEM). Grey bars show 30-s time intervals used to calculate the average speed at 7% and 21% $O_2$. Right: the bar graph shows fold change in average speed at 21% $O_2$ compared to 7% $O_2$. N = 6–9 assays. (**D**) *qui-1* mutants lack pheromone-evoked $Ca^{2+}$ responses in ADL. Left: average GCaMP3 signal intensity (*F*) divided by baseline intensity (*F_0*) plotted over time. Shading shows SEM. Light blue rectangle indicates period of C9 pheromone stimulation. Right: bar graph quantifying pheromone-evoked $Ca^{2+}$ responses. $\Delta F/F_0$ (%) was computed from 5-s intervals before the C9 stimulus was removed (*F*) and 5s before C9 stimulus was presented (*F_0*), indicated by the grey bars. N = 8 (Control) and N = 7 (*qui-1*); both strains carry a *dbEx941[Δsre-1p::GCaMP3v500; unc-122p:rfp]* transgene (**E**) *qui-1* mutants appear to lose $O_2$-evoked $Ca^{2+}$ responses in ADL. Left: $Ca^{2+}$ levels reported as a pseudo-ratio between the GCaMP6s and mKate2 fluorescence signals. Both proteins are expressed under the ADL-specific promoter *srh-220p*. Grey horizontal bars show intervals (30 and 15 s) used for calculating $\Delta R/R_0$ (%) in the bar graph (right), which quantifies $O_2$-evoked $Ca^{2+}$ responses in ADL. N = 15 (Control), N = 13 (*qui-1*); both strains carry a *dbEx1149[srh-220p:GCaMP6s(Ce):mKate2; lin-44p:gfp]* transgene. Shading shows SEM. Statistics: *p value ≤0.05, **p value ≤0.01, ***p value ≤0.001, ****p value ≤0.0001, Mann–Whitney *U*-test. Comparisons are with wild-type (WT) or Control, which is N2 bearing the indicated transgene.

The online version of this article includes the following figure supplement(s) for figure 2:

**Figure supplement 1.** Conservation of NACHT/WD40 containing proteins.

simple increase in $O_2$-evoked $Ca^{2+}$ responses in ADL does not explain the increased ability of *qui-1* mutants to escape 21% $O_2$.

## Disrupting *qui-1* enhances neurosecretion in ADL sensory neurons

To further probe how disrupting *qui-1* alters ADL function, we monitored neurosecretion from this neural pair using a fluorescently tagged insulin-like peptide, DAF-28;;mCherry, specifically expressed in ADL using the *srh-220* promoter. In *C. elegans*, insulin-like peptides are secreted through dense-core vesicles (DCVs) and accumulate in scavenger cells called coelomocytes (*Fares and Greenwald, 2001*). Accumulation of fluorescently tagged insulin-like peptides in these cells provides a readout of neurosecretion (*Lee and Ashrafi, 2008*; *Sieburth et al., 2007*). Using this assay, we found a striking increase in ADL neurosecretion in *qui-1* mutants compared to control (*Figure 3A*). Expressing wild-type *qui-1* exclusively in ADL fully rescued this enhanced neurosecretion phenotype (*Figure 3A*). Increased insulin secretion levels cannot be explained by increased expression from the *srh-220* promoter: the fluorescent intensity of free mKate expressed from this promoter was not altered in *qui-1* mutants (*Figure 3—figure supplement 1A*). These data suggest that disrupting QUI-1 function in ADL enhances neurosecretion from these neurons.

Increased neurosecretion could reflect delivery of a larger number of DCVs to release sites. To ask if *qui-1* altered DCV trafficking, we tagged IDA-1, a DCV-associated protein, with GFP and expressed this fusion protein exclusively in ADL. ADL is highly polarized: its cell body projects a dendrite anteriorly, to the animal's nose, and an axon that bifurcates at the nerve ring into ventral and dorsal projections that form synapses with post-synaptic partners. As expected, IDA-1::GFP fluorescence was localized to small bright puncta along ADL axons and more diffusely in the ADL cell body (*Figure 3—figure supplement 1C* II–IV). No signal was detected in dendrites. To quantify possible differences, we measured how the intensity of IDA-1::GFP signal changes when *qui-1* is defective. *qui-1* mutants did not show gross differences in the axonal distribution of IDA-1::GFP (*Figure 3B*). To assess if more IDA-1::GFP was retained in the cell body in *qui-1* mutants, we compared fluorescence signals between *qui-1* and control but did not observe any differences (*Figure 3—figure supplement 1B*). Moreover, *qui-1* mutants did not show obviously altered ADL morphology (*Figure 3—figure supplement 1C* I–III). These data suggest that enhanced neurosecretion from ADL in *qui-1* mutants is not due to increased DCVs accumulation in axons but may reflect an increased rate of release.

## Absence of *qui-1* increases neurosecretion from ADL in response to $O_2$-circuit input

Why do *qui-1* mutants exhibit increased neurosecretion from ADL neurons? A simple hypothesis, prompted by the increased behavioural response of *qui-1* mutants to 21% $O_2$, is that enhanced ADL neurosecretion is due to stronger coupling to input from URX. The soluble guanylyl cyclase GCY-35 acts as the main oxygen molecular sensor: null mutations in *gcy-35* disrupt $O_2$-evoked responses both at the circuit and behavioural level (*Busch et al., 2012*; *Laurent et al., 2015*; *Zimmer et al., 2009*). Consistent with this, disrupting *gcy-35* almost completely abolished the enhanced neurosecretion of *qui-1* mutants (*Figure 3C*). Overexpressing wild-type GCY-35 in URX, using the *flp-8* or *gcy-32* promoters, rescued the ADL neurosecretion phenotype of *qui-1;gcy-35* double mutants, although not completely (*Figure 3D*). We conclude that increased neurosecretion from ADL neurons in *qui-1* mutants reflects an enhanced response to $O_2$ partly mediated by URX neurons.

Our experiments with *qui-1;gcy-35* double mutants predict that manipulating ambient $O_2$ levels should shape ADL neurosecretion in *qui-1* mutants. To investigate this hypothesis, we grew controls and *qui-1* mutants at 7% and 21% $O_2$ and assayed neurosecretion from ADL. In control animals ADL neurosecretion was unaffected by $O_2$ experience (*Figure 3E*). By contrast, neurosecretion from ADL was significantly modulated by $O_2$ experience in *qui-1* mutants (*Figure 3E*). Mutants kept at low $O_2$ concentrations showed markedly less ADL neurosecretion than animals kept at 21% $O_2$ (*Figure 3E*). Together, these data support the hypothesis that disrupting *qui-1* confers $O_2$-evoked neurosecretion on ADL neurons.

URX and ADL neurons are connected by gap junctions to RMG interneurons in the hub-and-spoke circuit (*Figure 1A*; *Cook et al., 2019*; *Macosko et al., 2009*). Signalling from the NPR-1 neuropeptide receptor in RMG modulates communication across the hub-and-spoke circuit. In the N2 genetic background, a hyperactive version of this neuropeptide receptor, NPR-1 215V, impedes communication

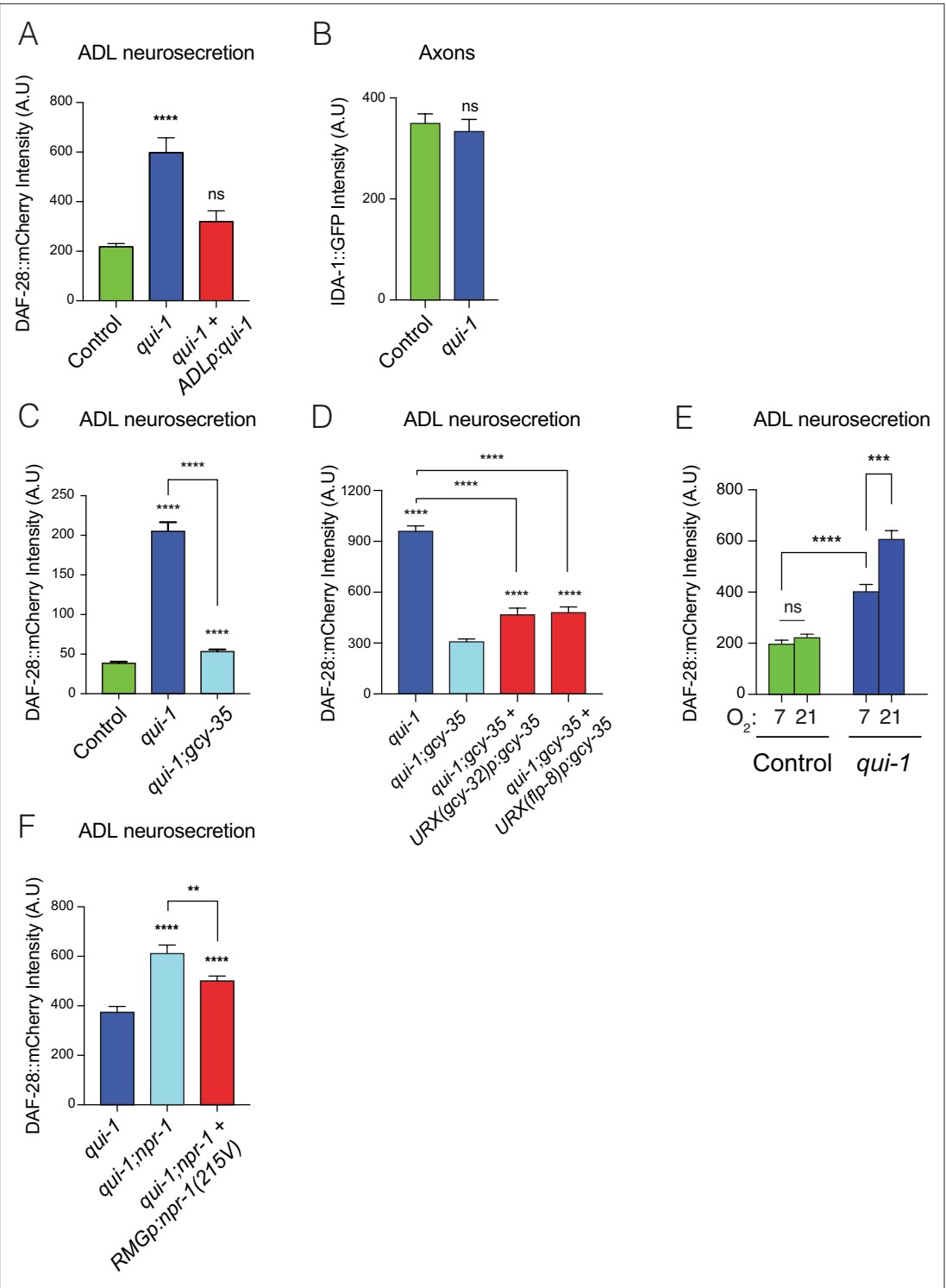

**Figure 3.** Loss of *qui-1* confers O$_2$-evoked neurosecretion on ADL sensory neurons. (**A**) Disrupting *qui-1* increases neurosecretion from ADL. This phenotype is rescued by expressing *qui-1* cDNA specifically in ADL (Δ*sre-1p*). N = 22 (Control), N = 16 (*qui-1*), N = 15 (*ADLp* rescue). (**B**) Loss of *qui-1* does not lead to increased axonal accumulation of dense-core vesicles (DCVs). Bar graph shows axonal levels of IDA-1::GFP, a DCV marker we expressed exclusively in ADL using *srh-220p*. N = 42 (Control), N = 34 (*qui-1*) both strains carry *dbEx1151[srh-220p:ida-1::gfp; srh-220p:mKate2; lin-44p:gfp]*. Increased ADL neurosecretion in *qui-1* mutants depends on the molecular O$_2$-sensor GCY-35 (**C**), which acts partly in URX neurons (**D**). (**C**) N = 21 (Control), N = 30 (*qui-1*), N = 27 (*qui-1; gcy-35*). (**D**) N = 60 (*qui-1*), N = 46 (*qui-1; gcy-35*), N = 46 (URX *gcy-32p* rescue), N = 32 (URX *flp-8p* rescue). (**E**) O$_2$ levels modulate ADL neurosecretion in *qui-1* mutants but not wild-type (WT). Animals were raised at either 7% or 21% O$_2$ from egg to young adult. N = 30 (Control, 7% O$_2$), N = 22 (Control, 21% O$_2$), N = 23 (*qui-1*, 7% O$_2$), N = 35 (*qui-1*, 21% O$_2$). (**F**) RMG signalling contributes to ADL neurosecretion

*Figure 3 continued on next page*

*Figure 3 continued*

in *qui-1* mutants. Rescue was achieved using two promoters that overlap only in RMG. The *flp-21* promoter drives a floxed transcriptional STOP signal followed by the *npr-1*(215V) isoform, *flp-21p:flox:STOP:flox:npr-1(215V)*; the *ncs-1* promoter drives the Cre recombinase (*ncs-1p:Cre*). N = 27 (*qui-1*), N = 22 (*qui-1;npr-1*), N = 33 (*RMGp* rescue). In A and C–F, bar graphs report the accumulation of DAF-28::mCherry fluorescence in coelomocytes following its release from ADL; all strains carry *ftIs25[srh-220p:daf-28::mCherry; myo-2p:gfp; unc-122p:gfp]*. Statistics: \*\*p value ≤0.01; \*\*\*p value ≤0.001; \*\*\*\*p value ≤0.0001; ns, not significant. Mann–Whitney *U*-test. Comparisons are against Control in A–C, against *qui-1;gcy-35* in D, and against *qui-1* in F. Control refers to N2 carrying the indicated transgene.

The online version of this article includes the following figure supplement(s) for figure 3:

**Figure supplement 1.** Enhanced secretion of DAF-28::mCherry from ADL reflects increased dense-core vesicle (DCV) release.

across the circuit (***Macosko et al., 2009***). To test if RMG activity alters ADL neurosecretion, we assayed *qui-1* and *qui-1;npr-1* double mutants. We observed higher levels of neurosecretion from ADL in *qui-1;npr-1* double mutants (***Figure 3F***). Expressing NPR-1 215V in RMG partially rescued the ADL phenotype of *qui-1;npr-1* double mutants (***Figure 3F***). We conclude that NPR-1 signalling in RMG neurons can suppress neurosecretion from ADL. Taken together, these and previous data suggest enhanced ADL neurosecretion in *qui-1* mutants is principally driven by increased ADL responsiveness to $O_2$ input from the hub-and-spoke circuit.

## Disrupting sensory perception in ADL increases its responsiveness to $O_2$ input

Is the increased coupling of ADL to the hub-and-spoke circuit specific to *qui-1* mutants or an adaptation to impaired sensory perception? A group of genes involved in sensory perception and associated with Bardet–Biedl syndrome, called *bbs* genes in *C. elegans*, has been proposed to reduce, by an unknown mechanism, neurosecretion (***Lee et al., 2011***). *bbs* genes encode components of a large protein complex involved in intraflagellar transport, the BBsome, which couples cargo vesicles to motor proteins for delivery to cilia. *bbs* mutants exhibit sensory defects, and, like *qui-1*, show increased DCV release from ADL (***Lee et al., 2011***). We asked if *bbs* mutants also show an increase in $O_2$-evoked behavioural responses. Of the five *bbs* mutants we studied, three, *bbs-1, -2,* and *-7*, responded to 7% $O_2$ by slowing down significantly more than N2; one, *bbs-7*, also showed increased activity at 21% $O_2$, behaving like *qui-1* (***Figure 4A*** and ***Figure 4—figure supplement 1A–C***). These data suggest more sensory defective mutants could display elevated $O_2$-evoked responses.

A search of the sequencing data from our mutant collection revealed two mutant strains carrying missense mutations in genes previously associated with impaired sensory perception, *wrt-6* (WaRThog, a hedgehog-related protein) and *fig-1* (dye-*F*illing abnormal, expressed *I*n *G*lia) (***Bacaj et al., 2008***; ***Hao et al., 2006***). The *wrt-6* (*db102*) allele substituted a conserved threonine residue (T460I) (***Figure 4—figure supplement 1D***) essential for autocleavage and activation of Hedgehog-like secreted proteins; *fig-1* (*db1239*) allele changed a cysteine in a C6 domain into a tyrosine (C1951Y) (***Figure 4—figure supplement 1E***). A wild-type copy of *wrt-6* entirely rescued the $O_2$-escape phenotype of *db102* mutants, confirming that this phenotype reflect loss of *wrt-6* function (***Figure 4B***). To test if defects in *fig-1* elevated $O_2$-escape behaviour, we assayed multiple *fig-1* loss-of-function alleles (***Figure 4—figure supplement 1E***). All *fig-1* mutants showed an increased $O_2$-response characterized by reduced locomotory activity at 7% $O_2$ (***Figure 4C***), suggesting that the absence of *fig-1* leads to stronger $O_2$-evoked responses. We also injected *fig-1* mutants with a wild-type copy of *fig-1* but failed to rescue $O_2$-escape behaviour (data not shown). Appropriate protein levels may be necessary for correct *fig-1* function.

Together, our data suggest a model in which compromising sensory input increases ADL's responsiveness to $O_2$ input from the hub-and-spoke circuit. To test this, we measured ADL neurosecretion in *wrt-6* and *fig-1* mutants and observed a robust increase in both mutants (***Figure 4D***). *wrt-6* and *fig-1* are expressed in glia and not neurons (***Bacaj et al., 2008***; ***Hao et al., 2006***) and are unlikely to regulate neurosecretion directly. We next asked if enhanced neurosecretion from ADL in sensory defective mutants depended on $O_2$ input. We raised *bbs-7* mutants, which showed the strongest $O_2$-evoked behavioural reponses among the sensory defective mutants we had studied, at 7% and 21% $O_2$ and measured ADL neurosecretion. *bbs-7* mutants grown at 7% $O_2$, when URX–RMG activity is low, lost their enhanced neurosecretion phenotype and showed secretion levels indistinguishable from control animals reared at 7% $O_2$ (***Figure 4E***). These data suggest that ADL neurons release more DCVs in

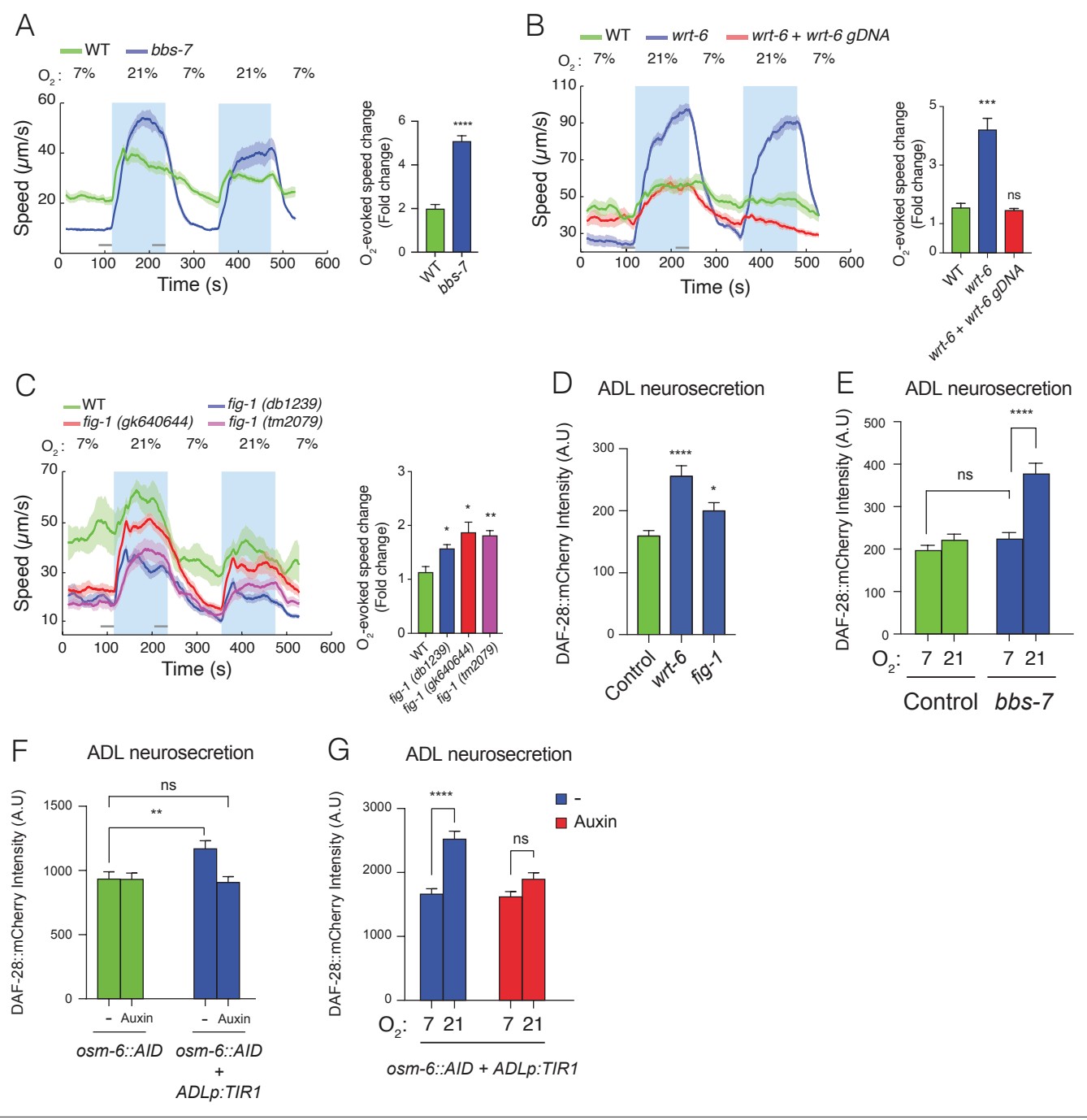

**Figure 4.** Disrupting sensory perception in ADL confers O₂-evoked neurosecretion. (**A**) *bbs-7* mutants, which lack a subunit of the BBSome complex and have impaired cilia formation and function, show increased behavioural responses to O₂ (see also **Figure 4—figure supplement 1**). (**B**) *wrt-6* mutants, which are defective in a hedgehog-related gene expressed in the glia surrounding the sensory endings of chemosensory neurons, show increased O₂-evoked behaviours. The phenotype is rescued by a wild-type copy of *wrt-6*. (**C**) Null mutants of , *fig-1* another gene expressed in glia whose loss causes chemosensory defects, show increased O₂ responses similar to *fig-1* (*db1239*) allele isolated in our screen (see **Figure 4—figure supplement 1**). (**A–C**) Left: lines show average speed, shading represents standard error of the mean (SEM), and grey bars represent 30-s intervals used to calculate the average speed at 7% and 21% O₂. Right: bar graphs show fold change in average speed at 21% O₂ compared to 7% O₂. N = 6–9 assays. (**D**) *wrt-6* and *fig-1* mutants show increased neurosecretion from ADL. N = 22 (Control), N = 21 (*wrt-6*), N = 18 (*fig-1*). (**E**) Increased ADL neurosecretion in *bbs-7* mutants reflects increased responsiveness to O₂ stimuli. Animals experienced 7% or 21% O₂ from egg to young adult, as indicated. N = 30 (Control, 7% O₂), N = 22 (Control, 21%O₂), N = 25 (*bbs-7*, 7% O₂), N = 25 (*bbs-7*, 21% O₂). (**F**) Knocking down OSM-6, a protein essential for intraflagellar transport and cilia function, exclusively in ADL alters neurosecretion from this neuron. Animals were grown from egg to young adult on control plates (−) or plates containing 1 mM Auxin (Auxin). N = 35 (*osm-6::AID*, −), N = 40 (*osm-6::AID*, Auxin), N = 36 (*osm-6::AID* + ADLp:*TIR1*, −), N = 30 (*osm-6::AID* +

*Figure 4 continued on next page*

**Figure 4 continued**

ADLp:*TIR1*, Auxin). (**G**) ADL-specific knockdown of OSM-6::AID confers $O_2$-evoked neurosecretion. Animals were grown from egg to young adult on control plates (–) or plates containing 1 mM Auxin (Auxin) at either 7% or 21% $O_2$ as indicated. $N = 41$ (7%, –), $N = 38$ (21%, –), $N = 37$ (7%, Auxin), $N = 36$ (21%, Auxin). (**D–G**) Bar graphs show DAF-28::mCherry fluorescence accumulated in coelomocytes following release from ADL, and all strains carry *ftIs25[srh-220p:daf-28::mCherry; myo-2p:gfp; unc-122p:gfp]*. Statistics: *p value ≤0.05; **p value ≤0.01; ***p value ≤0.001; ****p value ≤0.0001; ns, not significant. Mann–Whitney *U*-test. Unless indicated, comparisons are against wild-type (WT) or Control, which is N2 carrying the indicated transgene.

The online version of this article includes the following figure supplement(s) for figure 4:

**Figure supplement 1.** Mutants defective in subunits of the BBSome complex show enhanced $O_2$ responses.

**Figure supplement 2.** Impairment of OSM-6::AID in ADL disrupts its cilia formation and results in a cell-specific dye filling defect.

---

*bbs*, *wrt-6*, and *fig-1* mutants than controls. For *bbs-7* and *qui-1*, and potentially for other sensory-defective mutants, enhanced ADL neurosecretion reflects increased responsiveness to $O_2$ input.

We next asked if disrupting sensory input into ADL changes its functional coupling to the hub-and-spoke circuit. We impaired sensory input into ADL by cell specifically disrupting the function of its cilia. Sensory cilia are necessary for ADL's chemosensory activity, and proper cilia formation is supported by OSM-6, an intraflagellar transport protein (*Collet et al., 1998*). *osm-6* mutants display truncated cilia, fail to take up the lipophilic dye DiO, and show severe chemosensory defects (*Hedgecock et al., 1985*; *Perkins et al., 1986*). To cell specifically disrupt OSM-6 in ADL, we introduced a sequence encoding an Auxin inducible degron (AID) (*Nishimura et al., 2009*; *Zhang et al., 2015*) in frame and just upstream of the stop codon of *osm-6* using CRISPR/Cas9. *osm-6::AID* knock-in animals did not show any defect compared to wild-type in their ability to take up DiO (*Figure 4—figure supplement 2A, B*), suggesting that OSM-6::AID is functional. ADL-restricted expression of the F-box protein TIR1, which selectively targets proteins containing the AID tag for degradation (*Nishimura et al., 2009*; *Zhang et al., 2015*), was sufficient to reduce ADL dye filling in *osm-6::AID* animals (*Figure 4—figure supplement 2C* III and *Figure 4—figure supplement 2D*). Adding Auxin further reduced ADL dye filling (*Figure 4—figure supplement 2C* IV and *Figure 4—figure supplement 2D*) compared to a control strain expressing TIR1 in ADL in the absence of the *osm-6::AID* allele. This is consistent with recent reports that TIR1 can target proteins for degradation in the absence of Auxin (*Hills-Muckey et al., 2021*). Taken together these data confirm cell-specific knockdown of OSM-6 and that increased knockdown progressively impairs ADL cilia integrity.

Cell-specific disruption of OSM-6::AID resulted in dye filling defects in ADL. To assess if this disruption was sufficient to heighten responsiveness to pre-synaptic $O_2$ input, we assayed ADL neurosecretion in *osm-6::AID* animals. *osm-6::AID* animals expressing TIR1 in ADL displayed elevated neurosecretion levels compared to *osm-6::AID* controls (*Figure 4F* and *Figure 4—figure supplement 2E*), whereas growing these animals in the presence of Auxin suppressed this increase (*Figure 4F*). To test if increased neurosecretion reflected enhanced responsiveness to $O_2$ input, we grew *osm-6::AID* animals expressing TIR1 in ADL at 7% $O_2$ or 21% $O_2$. Altering $O_2$ concentrations modulated ADL neurosecretion levels only in *osm-6::AID* animals expressing TIR1 grown in the absence of Auxin (*Figure 4G*), suggesting that limited impairment of OSM-6::AID in ADL confers $O_2$-evoked neurosecretion on this neuron pair. We conclude that limited impairing of OSM-6 function in ADL confers $O_2$-evoked neurosecretion. Taken together these data confirm that the enhanced coupling of ADL to the hub-and-spoke circuit observed in *qui-1* and other sensory defective mutants most likely results from cell autonomous sensory defects in ADL.

## Elevating NPR-22 expression in ADL underpins $O_2$-evoked neurosecretion

Defects in sensory perception remodels ADL properties to enhance neurosecretion in response to input from URX–RMG. To investigate the molecular details behind this process, we labelled ADL neurons by expressing mKate2 from an ADL-specific promoter (*srh-220*p), used fluorescence-activated cell sorting (FACS) to sort ADL from freshly dissociated wild-type controls and *qui-1* mutants, and then profiled the ADL transcriptome using RNAseq. Enrichment analysis highlighted ADL as the most enriched neural class in our dataset (*Figure 5—figure supplement 1A*). Our RNAseq data included known ADL-specific transcripts such as *srh-234* (*Gruner et al., 2014*) and *srh-279* (*Vidal et al., 2018*), and *qui-1* itself, but not transcripts expressed in neighbouring neurons such as ASK and ASI

(*Supplementary file 1* and data not shown). Consistent with its function as a chemosensory neuron, ADL expresses a large number of chemoreceptors (*Supplementary file 2*), as well as several neuropeptide receptors (*Supplementary file 3*) and neuropeptides (*Supplementary file 4*).

Principal component analysis confirmed that we could robustly differentiate *qui-1* from control samples (*Figure 5—figure supplement 1B*). We next examined genes differentially regulated between controls and *qui-1* mutants in ADL (*Supplementary file 5*). The majority of differentially regulated

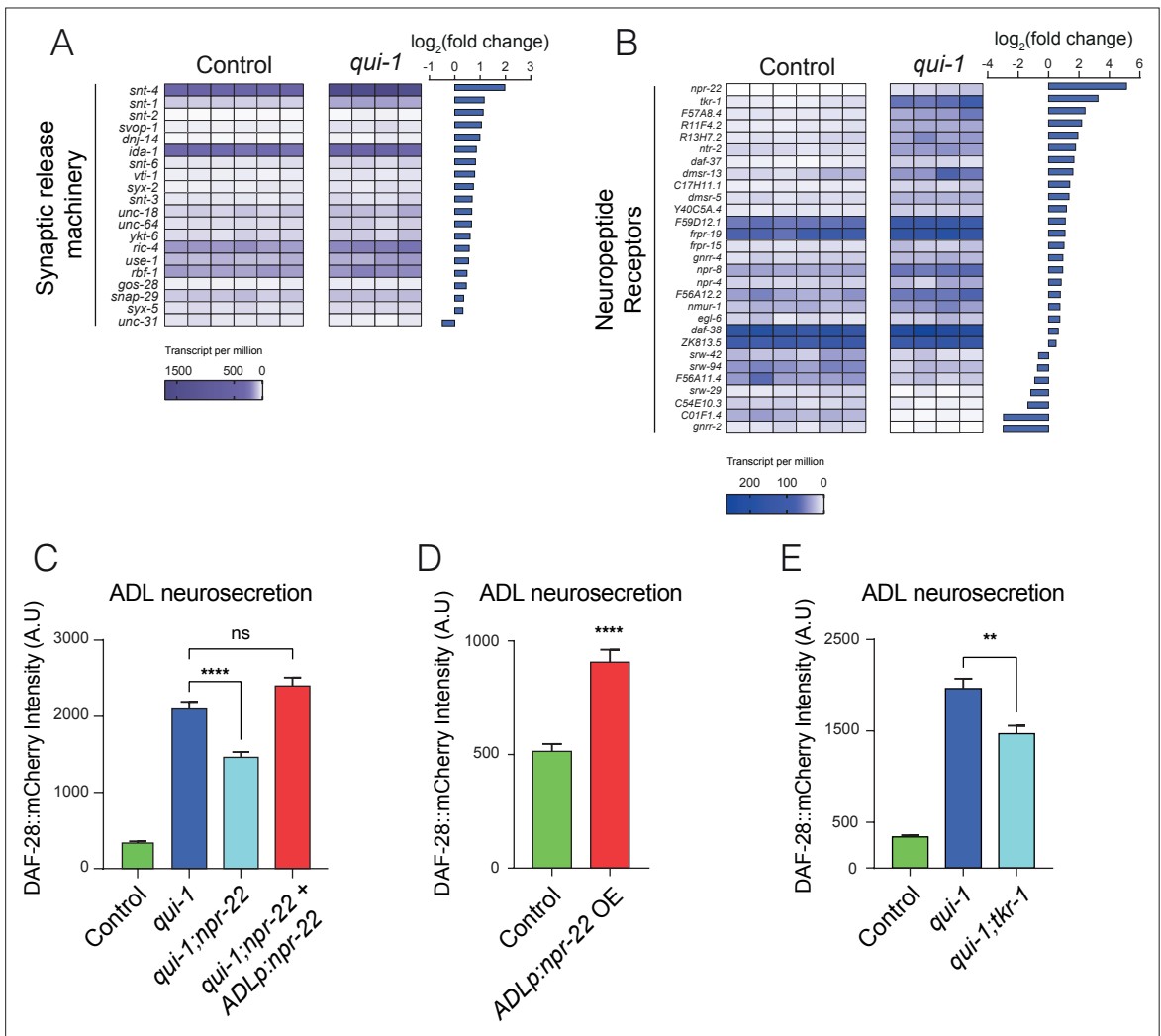

**Figure 5.** Increased $O_2$-evoked neurosecretion from ADL in *qui-1* mutants is associated with reconfigured peptidergic communication. (**A**) *qui-1* mutants upregulate a suite of genes that control synaptic and/or dense-core vesicle (DCV) release. (**B**) Loss of *qui-1* reprograms neuropeptide receptor expression in ADL. (**A, B**) Heat maps show expression values (transcript per million, tpm) for each gene across biological replicates, which are indicated by columns. To target analysis, we used lists generated in a review of the neural genome (*Hobert, 2013*), and selected the annotation 'Synaptic release machinery' (**A**) or 'Neuropeptide receptor' (**B**). All genes matching these criteria were included unless their expression was below 10 tpm in both genotypes or the *q*-value was >0.05 (see *Supplementary file 5*). Animals from Control and *qui-1* samples carry *dbIs47[srh-220p:mKate2; lin-44p:gfp]*. (**C**) The neuropeptide receptor NPR-22 promotes neurosecretion from ADL in *qui-1* mutants; this phenotype can be rescued by expressing *npr-22* cDNA (encoding NPR-22 isoform b) from an ADL-specific promoter (*srh-220p*). N = 33 (Control), N = 54 (*qui-1*), N = 53 (*qui-1;npr-22*), N = 52 (*ADLp* rescue). (**D**) Overexpressing the same *npr-22* cDNA construct specifically in ADL is sufficient to stimulate neurosecretion in control animals. N = 46 (Control), N = 44 (*ADLp* OE). (**E**) The tachykinin receptor *tkr-1* also stimulates ADL neurosecretion in *qui-1* mutants. N = 38 (Control), N = 37 (*qui-1*), N = 38 (*qui-1;tkr-1*). (**C–E**) Bar graphs show the intensity of DAF-28::mCherry accumulated in coelomocytes following release from ADL, and all strains carry *ftIs25[srh-220p:daf-28::mCherry; myo-2p:gfp; unc-122p:gfp]*. Statistics: **p value ≤0.01; ****p value ≤0.0001; ns, not significant. Mann–Whitney *U*-test. In C and E, comparisons are against *qui-1*, while in D comparisons are against Control, which is N2 carrying the indicated transgene.

The online version of this article includes the following figure supplement(s) for figure 5:

**Figure supplement 1.** Profiling of ADL sensory neurons.

**Figure supplement 2.** Disrupting *npr-22*, *tkr-1*, or synaptic communication is not sufficient to reduce the $O_2$-escape behaviour of *qui-1* mutants.

genes were strongly upregulated in mutant samples (*Figure 5—figure supplement 1C, D*). When we selected all known genes associated with DCV release (*Hobert, 2013*) that were also differentially regulated, almost all showed elevated expression in *qui-1* mutants (*Figure 5A*). *qui-1* mutants also showed altered chemosensory receptor levels: more than half of all the chemoreceptors expressed in ADL were differentially regulated (*Figure 5—figure supplement 1E, F*). These data suggest that loss of *qui-1* extensively remodels ADL gene expression. It is important to note that while we cannot completely exclude the possibility that background mutations could contribute to some of the gene expression changes we observe, we used strains that were extensively outcrossed (see Methods) in our experiments, giving us confidence the contribution of background mutations should be minimal.

Our data indicate that in *qui-1* mutants ADL responds more strongly to $O_2$ input from the hub-and-spoke circuit by releasing more DCVs. The absence of increased $O_2$-evoked $Ca^{2+}$ responses in ADL in *qui-1* animals (*Figure 2E*) argues against elevated $Ca^{2+}$ being responsible for increased $O_2$-evoked DCV release. Other second messengers, for example cyclic adenosine monophosphate (cAMP), can strongly stimulate DCVs release (*Steuer Costa et al., 2017*). We hypothesized that altered G-protein-coupled receptor signalling in ADL could account for $O_2$-evoked DCV release in these neurons. Several neuropeptide receptors were differentially expressed in ADL between control and *qui-1* (*Figure 5B*). Most prominent of these was neuropeptide receptor 22 (*npr-22*), which was one of the most highly upregulated genes in *qui-1* mutants compared to control (*Figure 5—figure supplement 1D*). To investigate if elevated NPR-22 levels in *qui-1* mutants explained the increased DCV release in ADL, we compared ADL neurosecretion in *qui-1* and *qui-1;npr-22* animals. The double mutant showed significantly reduced neurosecretion compared to *qui-1* mutants (*Figure 5C*). This phenotype was completely rescued by selectively expressing *npr-22* in ADL, confirming that *npr-22* is necessary to sustain ADL's higher neurosecretion levels in *qui-1* mutants, and acts in ADL itself. *npr-22* is not expressed at appreciable levels in wild-type ADL according to our profiling data (*Supplementary file 5*). To test if inducing *npr-22* expression is sufficient to stimulate ADL neurosecretion, we overexpressed the neuropeptide receptor in wild-type ADL neurons. Increasing *npr-22* expression was sufficient to induce a higher rate of neurosecretion from ADL (*Figure 5D*). These data suggest increased peptidergic signalling through NPR-22 is necessary and sufficient to promote ADL neurosecretion.

Disrupting *npr-22* did not completely suppress the neurosecretion phenotype of *qui-1* mutants. We investigated if other neuropeptide receptors whose expression in ADL was induced in *qui-1* mutants augmented $O_2$-evoked neurosecretion from ADL. The second such neuropeptide receptor gene in our list (*Figure 5B*) was the TachyKinin Receptor 1, *tkr-1*. Loss of *tkr-1* also significantly decreased neurosecretion from ADL to levels comparable to those in *qui-1;npr-22* (*Figure 5E*).

Is disrupting either *npr-22* or *tkr-1* sufficient to suppress $O_2$-evoked escape behaviour in *qui-1* mutants? Both *qui-1;npr-22* and *qui-1;tkr-1* animals show $O_2$-escape responses similar to those of *qui-1* mutants (*Figure 5—figure supplement 2A, B*), and overexpressing *npr-22* in ADL neurons did not confer increased $O_2$-escape behaviour compared to wild-type animals (*Figure 5—figure supplement 2C*). This suggests that while increased expression of *npr-22* and *tkr-1* underpin increased ADL neurosecretion in *qui-1* mutants, they are not sufficient to explain the increased $O_2$-escape behaviour in these animals. To further address if neurosecretion from ADL is required for the $O_2$-escape behaviour of *qui-1* mutants, we expressed tetanus toxin (TeTX) in *qui-1* mutants. TeTX is predicted to cleave SNB-1, the main synaptobrevin expressed in ADL neurons. The $O_2$-escape behaviour of *qui-1* animals expressing TeTX in ADL was similar to that of *qui-1* mutants (*Figure 5—figure supplement 2D*), suggesting that ADL neurosecretion is not essential for the increased $O_2$-escape response of *qui-1* mutants. Whether this reflects redundancy between *qui-1*-expressing neurons, and/or a role for altered ADL communication via gap junctions, is still unclear. We conclude that disrupting sensory responsiveness in ADL, by the *qui-1* mutation, increases the coupling of this neuron to the hub-and-spoke circuit by reconfiguring the expression of neuropeptide receptors that facilitate DCVs release.

## Discussion

Cross-modal plasticity is thought to involve recruitment of impaired neurons to process additional sensory modalities. The molecular details of such rearrangement are not yet clear. Here, we use forward genetics as an entry point to seek mechanisms that increase *C. elegans'* responsiveness to an oxygen ($O_2$) sensory cue. Several of the mutants we identify simultaneously increase responsiveness to $O_2$ while disrupting other sensory responses – a hallmark of cross-modal plasticity. We analyse one of

these mutants, *qui-1*, which is defective in the ortholog of mammalian NWD1, in depth. We show that loss of QUI-1 prevents the ADL sensory neurons from responding to pheromone, but increases ADL neurosecretion in response to input from upstream $O_2$-sensing neurons. Loss of *qui-1* thus recruits ADL sensory neurons more strongly into the $O_2$-sensing circuit. We observe a similar change in additional sensory defective animals as well as in animals with ADL-specific impairment of the intraflagellar transport protein OSM-6. Loss of *qui-1* is associated with extensive changes in gene expression in ADL neurons, notably induced expression of neuropeptide receptors, including NPR-22 and TKR-1. Elevated expression of NPR-22 and TKR-1 increase the coupling of ADL neurosecretion to $O_2$ input in *qui-1* mutants. We propose that impairing sensory perception can sensitize sensory neurons to other sensory modalities by reconfiguring peptidergic circuits.

ADL-specific expression of *qui-1* rescues both the increased $O_2$-escape phenotype of *qui-1* mutants and enhanced neurosecretion from ADL (*Figures 2B and 3A*), indicating that *qui-1* acts cell autonomously to regulate neurosecretion. Impairing the primary $O_2$-sensing mechanism, by disrupting the molecular oxygen sensor GCY-35, restores ADL neurosecretion in *qui-1* mutants to levels observed in controls (*Figure 3C*), confirming that increased ADL neurosecretion in *qui-1* is driven primarily by activity originating outside ADL. Together with data showing that *qui-1* mutants exhibit normal $O_2$-evoked $Ca^{2+}$ responses in the primary $O_2$-sensing neurons URX, AQR, PQR, and in the hub interneurons RMG (*Figure 1—figure supplement 1C–F*), this suggests that in *qui-1* mutants ADL neurons are more sensitive to incoming pre-synaptic activity.

Selectively expressing *gcy-35* cDNA in the URX $O_2$ sensors partially rescues the ADL neurosecretion phenotype of *qui-1;gcy-35* mutants (*Figure 3D*), confirming that URX helps drive increased ADL neurosecretion in *qui-1* mutants. Disrupting the inhibitory neuropeptide receptor *npr-1* further increases ADL neurosecretion in *qui-1* mutants, and this phenotype is partially rescued by expressing *npr-1* cDNA specifically in RMG interneurons (*Figure 3F*). This further supports a model in which activity from the hub-and-spoke circuit propagates from URX–RMG to ADL to stimulate neurosecretion. Consistent with this, in *qui-1* mutants, but not in controls, ADL neurons show $O_2$-evoked neurosecretion: prolonged exposure to low (7%) or high (21%) $O_2$ concentrations modulates ADL neurosecretion (*Figure 3E*). These data confirm that disrupting *qui-1* recruits ADL more strongly into the hub-and-spoke circuit. It still remains unclear how this increased coupling leads to enhanced $O_2$-escape behaviour. Blocking ADL's output by expressing TeTX has no effect on the $O_2$-escape behaviour of *qui-1* mutants. This negative result could reflect either redundancy between ADL and other *qui-1*-expressing neurons, or a still unidentified molecular mechanism, perhaps via gap junctions, by which ADL promotes the $O_2$ responses of this mutant.

Previous work in *C. elegans* has described cross-modal plasticity in a touch receptor/olfactory circuit paradigm (*Rabinowitch et al., 2016*). In this paradigm, worms with touch receptor defects show enhanced odorant responses compared to wild-type controls because activated touch receptors release an inhibitory neuropeptide, FLP-20, that downregulates communication between the AWC olfactory neurons and their post-synaptic target, the AIY interneurons. Loss of touch receptor function thus enhances odorant responses by disinhibiting AWC–AIY communication. In this mechanism, the defective touch receptors do not contribute to the enhanced odorant sensing. FLP-20 appears to act as a general arousal signal, relaying information about mechanical stimulation to multiple circuits (*Chew et al., 2018*). By contrast, in the paradigm we describe, the defective sensory neuron, ADL, becomes more strongly incorporated in the circuit mediating the enhanced modality, $O_2$ response.

Several questions remain outstanding. How does disrupting *qui-1*, and ADL sensory function, lead to extensive changes in ADL gene expression? Comparing gene expression in ADL between *qui-1* and wild-type controls reveals altered expression of several transcription factors, including members of the nuclear hormone receptor family (*nhr*), the *egl-46* zinc-finger protein (*Wu et al., 2001*) and the storkhead box protein *ham-1* (*Feng et al., 2013*). Some of these transcription factors show substantial (e.g. >30-fold) induction, and are orthologs of immediate early genes in mammals. These transcription factors may contribute to the transcriptional changes in ADL.

Previous work has shown that mutations in BBS-7, a conserved protein involved in trafficking of molecular cargos along the primary cilium of neurons, and linked to Bardet–Biedl syndrome (*Liu and Lechtreck, 2018*; *Tan et al., 2007*), lead to increased neurosecretion from ADL (*Lee et al., 2011*). Mutations in *bbs-7* cause defects in cilia formation and in sensory perception. Our data suggest that increased neurosecretion from ADL in *bbs* mutants may reflect increased coupling to the $O_2$ circuit

(*Figure 4E*). Interestingly, *bbs-7* mutants show a range of other physiological phenotypes including small body size and delays in development (*Lee et al., 2011*; *Mok et al., 2011*). Both of these phenotypes can be suppressed by mutating *gcy-35* (*Mok et al., 2011*), consistent with them resulting from enhanced responses to $O_2$.

We identify several sensory defective mutants that display both enhanced neurosecretion from ADL and increased $O_2$-evoked locomotory responses. This correlation suggests cell-specific sensory defects in ADL could increase $O_2$ responsiveness. We test this prediction by knocking down OSM-6, an intraflagellar transport protein essential for correct cilium assembly, exclusively in ADL, resulting in a cell-specific reduction in dye filling (*Figure 4—figure supplement 2D*), a proxy for aberrant cilia formation and sensory defects (*Collet et al., 1998*; *Hilliard et al., 2004*; *Inglis et al., 2007*; *Perkins et al., 1986*). Cell-specific sensory defects in ADL result in heightened responsiveness to $O_2$ inputs which support an increased ADL neurosecretion (*Figure 4G*). It is important to note that a further reduction in OSM-6 levels and ADL's dye filling capacity, achieved by cultivating animals overexpressing TIR1 on Auxin, suppressed ADL's $O_2$-evoked neurosecretion. The TIR1 system used in this study makes it difficult to assess OSM-6::AID levels in the absence or presence of Auxin, however we speculate that while mildly disrupting ADL's cilium function increases ADL responsiveness to $O_2$ inputs, a greater impairment does not result in the same alteration. These data illustrate a repurposing of ADL sensory neurons in response to cell-specific sensory defects. This is further supported by the fact that both *fig-1* and *wrt-6* display increased ADL neurosecretion and an enhanced $O_2$ response, despite

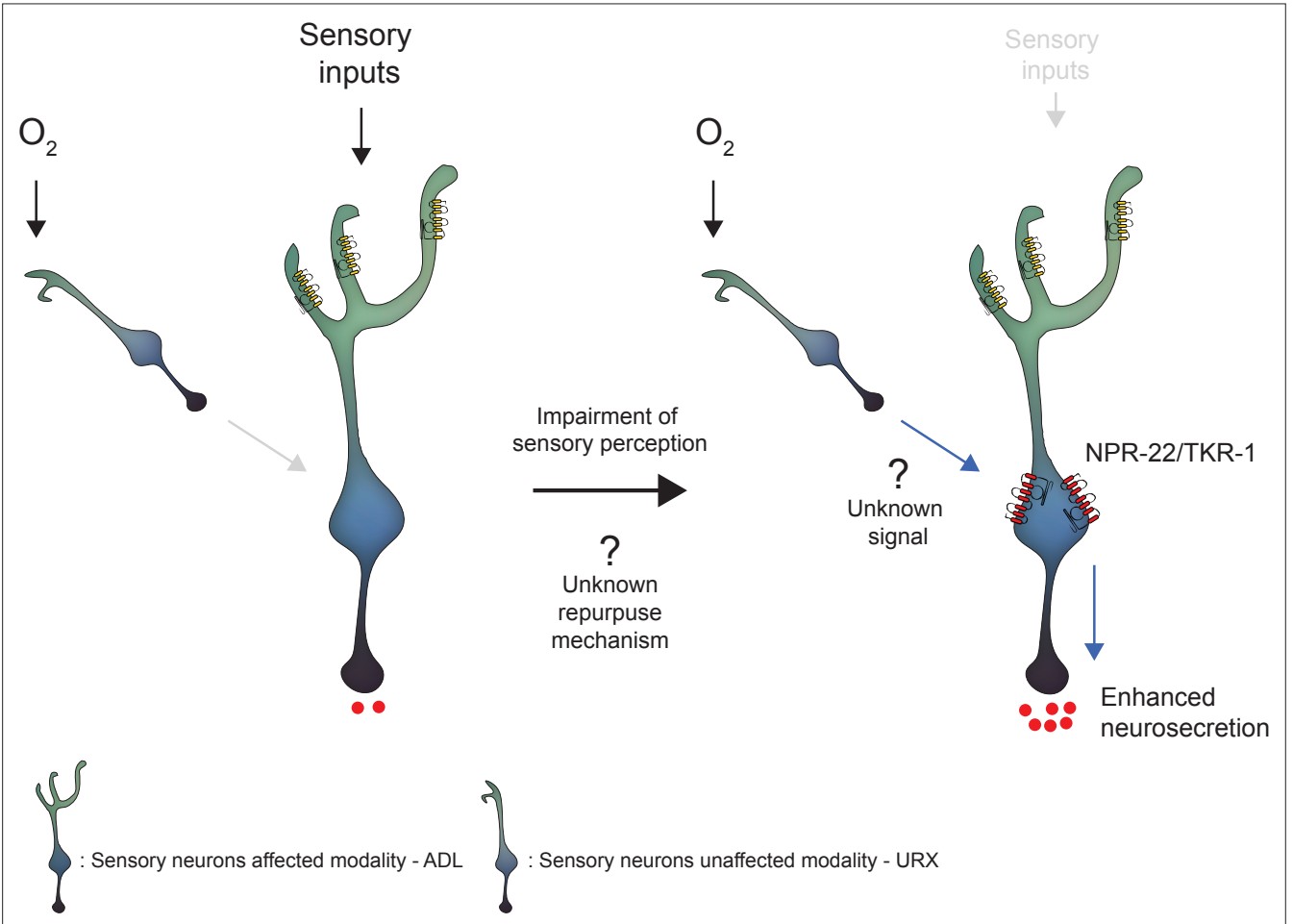

**Figure 6.** Model depicting the effect of disrupting sensory perception on ADL sensory neurons. Schematic depiction of model explaining ADL $O_2$-evoked neurosecretion in sensory defective mutants. When ADL sensory perception is functioning (left side), $O_2$ does not modulate ADL neurosecretion in wild-type, despite ADL receiving signals from $O_2$ sensors, including URX. When sensory perception is impaired (right side). ADL undertakes a transcriptional reconfiguration that results in stronger coupling of ADL neurosecretion to input from URX and other $O_2$ sensors. Much of the increase in $O_2$-evoked neurosecretion is conferred by increased expression of two receptors, NPR-22 and TKR-1, in ADL itself.

both genes being expressed in glial cells, suggesting their effect on ADL is indirect. Glia are required for proper cilia development and correct sensory perception, and both glia-ablated animals and *fig-1* mutants show defective responses to cues sensed by ADL (*Bacaj et al., 2008*). *wrt-6* mutants also exhibit a dye filling defect (data not shown), suggesting impairment in ADL's ability to sense external cues. These data taken together suggest that the enhanced responsiveness to $O_2$ input we observe in sensory defective mutants, characterized by increased $O_2$-evoked neurosecretion from ADL, is consistent with cell-specific defects in ADL sensory perception. Enhanced ADL neuroendocrine output may modulate aspects of the animal's physiology and behaviour we have not probed here.

Conceptually, our findings resonate with studies in vertebrates which find that loss of a sensory modality can lead to recruitment of input-deprived sensory cortex to process information from spared senses (*Lee and Whitt, 2015*; *Petrus et al., 2014*; *Rauschecker, 1995*). In the nervous system of *C. elegans*, the defective ADL sensory neurons becomes sensitized to pre-synaptic input associated with a different modality, $O_2$ sensing (*Figure 6*). This recruitment is supported by a transcriptional program in the sensory defective ADL that induces the expression of neuropeptide receptors including NPR-22 and TKR-1 (*Figure 6*). We speculate that reconfigured peptidergic circuits may be a common feature of cross-modal recruitment.

## Materials and methods

### Strains

*C. elegans* were grown at room temperature under standard conditions (*Brenner, 1974*). All assays used young adults (<24 hr old). In assays where Auxin treatment was used, animals were grown from eggs to young adult on 1 mM Auxin plates prepared as previously reported (*Zhang et al., 2015*). Transgenic animals were obtained by injecting DNA mixtures of an expression construct or fosmid and a co-injection marker, each at 20–40 ng/µl. A list of strains used is provided in *Supplementary file 6*. *E. coli* OP50 cultures were grown in 2xTY broth which was used to seed NGM plates.

### Mutagenesis

Animals were mutagenized with a 50 mM solution of ethylmethane sulfonate in M9 buffer (*Brenner, 1974*). To isolate mutants that preferentially aggregated on thick food we placed an ~0.2 × 0.2 cm patch of concentrated *E. coli* OP50 at the center of a much thinner circular lawn of OP50, ~5 cm in diameter, seeded on a 9-cm NGM dish. F2 progeny of mutagenized animals were washed 2× in M9 buffer and kept without food for ~30 min before being pipetted outside the thin bacterial lawn. Test experiments showed that under these conditions animals from non-aggregating strains strongly inhibited movement upon encountering the thin lawn. By contrast, individuals from aggregating strains continued moving quickly on the thin lawn but settled when they encountered the thick bacterial patch (*Figure 1B* – Step 1). Potential aggregating mutants were collected from the thick bacterial patch ~60 min after animals were added on the plate. These animals were then individually placed on a seeded NGM plate and their progeny scored for aggregation behaviour. Mutant lines showing an enhanced $O_2$-escape behaviour were outcrossed four times with the N2 laboratory strain to remove background mutations. Genomic DNA from outcrossed mutant lines was used for whole-genome sequencing, while outcrossed mutants were used for subsequent experiments.

### Behavioural assays

#### Aggregation assays

Assays were performed as described (*de Bono and Bargmann, 1998*). Sixty young adults were picked onto assay plates seeded 2 days earlier with 200 µl OP50. Animals were left undisturbed for 3 hr and scored as aggregating if they were in a group of 3 or more individuals in contact over >50% of their body length. Animals were considered to be at the lawn border if they were within 2 mm of the lawn edge. Scoring was performed blind to genotype. For each biological replicate:

$$\% \ Animals \ in \ clumps = \frac{number \ animals \ in \ clumps}{total \ number \ animals} * 100$$

$$\% \ Animals \ in \ clumps = \frac{number \ animals \ in \ clumps}{total \ number \ animals} * 100$$

## Locomotion assays

Assays were performed as described previously (*Laurent et al., 2015*). Low peptone NGM plates (0.13%, wt/vol bactopeptone) were seeded with 60 µl of OP50 broth 2 days before the assay. On the day of the assay, test plates were prepared by removing the edge of the bacterial lawn using a rubber stamp. Around 20 young adults were picked onto the lawn and left undisturbed for 10 min before starting the assay. A PDMS chamber was placed on top of the bacterial lawn and defined gas mixtures delivered to the chamber at 1.25 ml/min using a pump (PHD 2000, Harvard Apparatus). Worms were allowed to adapt to 7% $O_2$ for 2 min before videorecording started. Worms were recorded for 9 min while the gas mixture pumped into the chamber was changed from 7% to 21% $O_2$ every 2 min. Video-recordings were acquired at 2 frames per second (fps) using a Grasshopper camera (Point Grey) mounted on a stereomicroscope (Leica MZ6 and MZ7.5). Videos were analysed and animal speed calculated using a custom-written MATLAB software (Zentracker: https://github.com/wormtracker/zentracker; RRID SCR_022006 ). Average speed values were extracted using Metaverage, a custom-written MATLAB software. Bar graphs show a ratio of the average speed 30s before the end of the first 21% $O_2$ stimulus and the average speed 30-s before $O_2$ levels were first switched from 7% to 21% $O_2$. This ratio was computed for all biological replicates (independent assays) and entered into Prism for statistical analysis.

We plot the ratio of animal speed at 21% and 7% $O_2$, because it is a good proxy for the functionality of the $O_2$-sensing circuit in regulating locomotion. Animals that respond strongly to $O_2$, including natural wild isolates, *npr-1* null mutants, and the mutant strains we study here, differ from strains that respond poorly to $O_2$ such as N2 and *npr-1;gcy-35* in their locomotory responses to both 21% and 7% $O_2$. Animals with a functional $O_2$-sensing circuit become highly active at 21% $O_2$ and strongly quiescent at 7% $O_2$. Animals with a defective $O_2$ circuit move at intermediate speeds, with little change in their locomotory activity when $O_2$ levels change from 7% to 21%. The higher baseline speed at 7% $O_2$ of strains with a defective $O_2$-sensing circuit reflects these animals being adapted to the low activity of this circuit. *npr-1* animals kept for long periods at 7% $O_2$ also gradually begin to move faster at this $O_2$ concentration, and in fact respond more strongly than *npr-1* animals kept at 21% $O_2$ if $O_2$ levels rise.

## Molecular biology

### DNA extraction for whole-genome sequencing

Genomic DNA for whole-genome sequencing was isolated from 5 to 10 crowded 5-cm NGM plates. Animals were washed off plates in M9 buffer, rinsed 2× in M9 buffer to remove OP50, and frozen at −80°C. Genomic DNA was extracted from thawed samples using the DNeasy Blood and Tissue Kit (Qiagen). Samples were left in Lysis buffer (Buffer AL) for 3 hr at 56°C and DNA isolated following the manufacturer's instructions.

### Library preparation

Libraries for whole-genome sequencing were prepared using the Nextera XT DNA Library kit (Illumina) following the manufacturer's instructions. Library quality was checked on a Bioanalyzer using Agilent High Sensitivity Gel. Library concentration was assessed using KAPA Library Quantification Kits for Illumina (KAPA Biosystems) prior to sequencing on the Illumina HiSeq 4000 platform.

RNAseq of isolated neurons was adapted from *Picelli et al., 2014*. Briefly, fluorescently labelled neurons collected by FACS were lysed in 10 µl of 0.2% Triton X-100 (vol/vol) and 2 U/µl Rnase inhibitors. The reverse transcription reaction volumes were adjusted to 10 µl input and cDNA prepared using oligo dT primers and template-switching oligos (TSO) to enrich for polyadenylated transcripts and allow for pre-amplification of cDNA. cDNA was pre-amplified using custom PCR primers. 50 µl of PCR product from the pre-amplification step were purified using 50 µl of Ampure XP beads (Beckman Coulter), resuspended, and used at a concentration of 0.2 ng/µl as input for library preparation using the Nextera XT DNA Kit. Library preparation followed the manufacturer's instruction. The quality of RNAseq libraries was assessed on a Bioanalyzer (Agilent) using High Sensitivity Gels (Agilent). Library concentration was measured using a Qubit dsDNA High Sensitivity Kit (Thermo Fisher Scientific). Libraries were sequenced on the Illumina HiSeq 4000 platform.

## Ca²⁺ imaging

### Ca²⁺ imaging of O₂-evoked URX, AQR, and PQR activity

Five to ten young adult transgenic animals (<24 hr old) expressing the YC2.60 (URX) or the YC3.60 (AQR and PQR) Ca²⁺ sensors were glued to agarose pads (2% in M9 buffer, 1 mM CaCl2) using Dermabond tissue adhesive, with their body immersed in OP50 washed off from a seeded plate using M9. The animals were quickly covered with a PDMS microfluidic chamber and 7% O₂ pumped into the chamber for 2 min before imaging, to allow animals to adjust to the new conditions. Neural activity was recorded for 6 min with switches in O₂ concentration every 2 min. Imaging was on an AZ100 microscope (Nikon) equipped with a TwinCam adaptor (Cairn Research), two ORCAFlash4.0 V2 digital cameras (Hamamatsu), and an AZ Plan Fluor 2x objective with 2x zoom. Recordings were at 2 frame-per-second (fps) with a 500ms exposure time. Excitation light from a C-HGFI Intensilight lamp (Nikon) was passed through a 438/24 nm filter and an FF458-DiO₂ dichroic (Semrock). Emitted light was passed to a DC/T510LPXRXTUf2 dichroic filter in the TwinCam adaptor cube and then through 483/32 nm (CFP) or 542/27 nm (YFP) filters before collection on the cameras.

### Ca²⁺ imaging of O₂-evoked RMG activity

The imaging protocol was performed as reported for URX, except that *db104* mutants and matched controls were imaged on an Axiovert 200 microscope (Zeiss) with a 40x NA 1.2 C-Apochromat objective using an EMCCD Evolve 512 Delta camera (Photometrics), which gave higher signal-to-noise than the Nikon AZ100.

### Ca²⁺ imaging of pheromone-evoked ADL activity

We used olfactory chips (Microkosmos LLC, Michigan, USA) to image young transgenic adults (<24 hr old) expressing the GCaMP3 Ca²⁺ sensor specifically in ADL, as previously described (*Chronis et al., 2007*; *Jang et al., 2012*). Animals were kept under a constant flow of M13 buffer and after 2 min stimulated for 20s with C9 pheromone (10 nM in M13 buffer). Ca²⁺ imaging used a 40x NA 1.2 C-Apochromat lens on an Axiovert 200 microscope (Zeiss) equipped with a Dual View emission splitter (Photometrics) and an Evolve 512 Delta EMCCD camera (Photometrics). Acquisition was at 2 frame-per-second (fps) with a 100ms exposure. Excitation light was from a Lambda DG-4 (Sutter Instruments) and was passed through an excitation filter (AmCyan,Chroma), and a dichroic filter for GCaMP and RFP. A beam splitter (Optical Insights) was used to separate the GCaMP and RFP signal using a dichroic filter 514/30–25 nm (GFP) and 641 nm (RFP) (Semrock).

### Ca²⁺ imaging of O₂-evoked ADL activity

We imaged young transgenic adults (<24 hr old) co-expressing GCaMP6s and mKate2 from the ADL-specific *srh-220p*, in a bi-cistronic construct. Animals were immobilized with Dermabond glue, placed under a PDMS chamber and imaged on the same microscope and imaging setup used to image pheromone-evoked Ca²⁺ in ADL. Prior to recording activity, animals were pre-stimulated for 3 min to extinguish light-evoked ADL responses. Acquisition was at 2 frame-per-second (fps) with a 100ms exposure. O₂ concentration was switched between 7% and 21% O₂ every 2 min.

All recordings were analysed using Neuron Analyser, a custom-written MATLAB program available at https://github.com/neuronanalyser/neuronanalyser (RRID SCR_022007; copy archived at swh:1:rev:dcee5c20a60ec338010fb2ce0f52aeca725c75ba).

### Analysis

Bar graphs showing $\frac{\Delta R}{R_0}(\%)$ used YFP/CFP values extracted using Metaverage, a custom-written MATLAB software. YFP/CFP values at 7% O₂ taken 30-s before the first 21% O₂ stimulus (baseline) were subtracted from YFP/CFP values 30-s before the end of the first 21% O₂ stimulus (stimulus). This ratio was normalized by dividing with the baseline. Values calculated for each biological replicate were entered into Prism for statistical analysis. For ADL pheromone responses, average GCaMP3 intensity was calculated 5-s before the C9 stimulus was removed (*F*) and 5s before the C9 stimulus was presented (*F₀*). ADL pheromone-evoked Ca²⁺ responses were calculated as $\frac{\Delta F}{F_0}(\%)$. For ADL O₂-evoked responses, a pseudo-ratio of GCaMP6s over mKate2 signal was computed to account for changes in GCaMP6s intensity due to animal movement. O₂-evoked Ca²⁺ responses were calculated using the

average GCaMP6s/mKate2 signal for a 15-s window centered around the peak of the response and a 30-s window before stimulation with 21% $O_2$ ($R_0$) and expressed as $\frac{\Delta R}{R_0}(\%)$.

## Cell isolation and FACS

### Neuron isolation

Synchronized transgenic young adults in which the ADL neurons were specifically labelled using *srh-220p:mKate2* were washed 5× in M9 buffer to remove bacteria and then dissociated as described (*Beets et al., 2020*; *Kaletsky et al., 2018*). Briefly, animals were incubated for 6.5 min in Lysis buffer (200 mM Dithiothreitol (DTT), 0.25% Sodium dodecyl sulfate (SDS), 20 mM HEPES buffer (4-(2-hydroxyethyl)-1-piperazineethanesulfonic acid), 3% sucrose) washed 5× in M9 buffer, resuspended in 500 µl of 20 mg/µl Pronase in water, and pipetted up and down for 12 min at room temperature. The reaction was stopped by adding 250 µl of 2% FBS in PBS. Cells were filtered through a 5-µm syringe filter to remove clumps. mKate(+) cells were sorted using a Synergy High Speed Cell Sorter (Sony Biotechnology) with gates set using a negative control prepared in parallel from dissociated unlabelled N2 animals. Positive cells were collected into 10 µl of Triton X-100 0.2% (vol/vol) supplemented with 2 U/ml RNase inhibitors. Between 700 and 3000 cells were collected for each biological replicate.

## Microscopy

### ADL neurosecretion assay

We quantified neurosecretion in young transgenic adults (<24 hr old) expressing DAF-28::mCherry specifically in ADL neurons (*srh-220p:daf-28::mCherry*) together with a coelomocyte marker (*unc-122p:gfp*). To assay the effects of $O_2$ levels on ADL neurosecretion we grew animals form egg to young adult either at ambient $O_2$ (21% $O_2$) or at 7% $O_2$ using a hypoxic chamber ($O_2$ Control InVitro Glove Box, Coy Laboratories). To quantify neurosecretion, we imaged the most anterior pair of coelomocytes on a TE-2000 (Nikon) or a Ti2 (Nikon) wide-field microscope using a 10x air lens. Z-stack images were taken at subsaturating exposure for both GFP and mCherry intensities and analysed using ImageJ. Coelomocytes were delineated using the GFP signal and the mCherry signal measured in the same area. Values were plotted as arbitrary units of intensity.

### *srh-220p* validation

To validate the *srh-220* promoter construct we imaged wild-type and *qui-1* mutants carrying a transgene expressing mKate under the control of the *srh-220* promoter (*srh-220p:mKate*). We imaged the ADL cell body using a Ti2 (Nikon) wide-field microscope using a 40x air lens. Z-stacks were taken without saturating the mKate signal. The boundary of the ADL cell body was taken and intensities extracted using ImageJ. Data were plotted using Prism.

### DCV localization

To image DCVs the coding sequence of IDA-1, a DCV marker, was fused to GFP and expressed in the ADL pair of neurons using the ADL-specific promoter *srh-220*p (*srh-220p:ida-1::gfp*). Simultaneously, we specifically highlighted ADL by expressing cytosolic mKate (*srh-220p:mKate2*). Young adult double transgenic animals were imaged on a Ti2 (Nikon) wide-field microscope using a 40x air lens. Z-stack images were taken at subsaturating exposure for both GFP and mKate. We delineated the boundaries for the cell body and axon of ADL using the mKate signal, and measured signals in mKate+ pixels in the GFP channel using ImageJ. Values were plotted in Prism, in arbitrary units.

### QUI-1 expression and localization

To examine the expression and subcellular localization of *qui-1* we knocked in DNA encoding mNeonGreen in frame just upstream of the *qui-1* initiation codon. We imaged young adult hermaphrodites using a Ti2 (Nikon) microscope equipped with a DragonFly (Andor) spinning disk module and an EMCCD camera (iXon, Andor) with 40x or 60x objectives. Z-stacks of images acquired with subsaturating exposure times were analysed using ImageJ.

### ADL and amphid neurons dye filling

The ability of amphid neurons, including ADL, to take up the lipophilic dye DiO, was used as a proxy to monitor OSM-6::AID functionality. Briefly, worms were incubated with the DiO dye (10 μg/ml in M9 buffer) for 3 hr, and then transferred to a fresh plate for 1 hr to remove excess dye. To monitor amphid neuron dye filling, animals were inspected under a stereomicroscope (M165 FC, Leica). To monitor ADL dye filling in detail, animals were imaged on a Ti2 (Nikon) wide-field microscope using a 40x objective. Z-stack images were taken at subsaturating exposure for both DiO, imaged with a standard GFP filter, and tagBFP. To quantify ADL dye filling, the boundary of ADL cell body was taken using the tagBFP marker driven but the ADL-specific promoter *srh-220*p and DiO intensities extracted using ImageJ. Data were plotted as arbitrary units (A.U.) using Prism.

## Analysis

### Whole-genome sequencing

Whole-genome sequence data were analysed using a custom Python script, Cross_filter (https://github.com/lmb-seq/cross_filter; RRID SCR_022008 v1). Briefly, reads were checked for quality and aligned to the *C. elegans* reference genome. Lists of mutations for each sequenced strain were then cross-referenced with a compiled list of background mutations to generate a list of strain-specific mutations.

### RNA-sequencing

RNAseq data quality was checked using FastQC 0.11.7, before and after adaptor clipping; trimming quality was controlled using trimmomatic 0.38. Cleaned data were used for gene quantification using Salmon 1.1.0, *C. elegans* transcriptome (EnsemblMetazoa: release 46) and *C. elegans* genome (WBcel235) as decoy. We performed differential gene expression analysis using tximport 1.14.2 and DEseq2 1.26.0. Output from these programs was imported into a custom-made R program (PEAT, https://github.com/lmb-seq/PEAT, RRID SCR_021691, v1) to visualize differentially expressed genes across genotypes. EnrichmentBrowser 2.16.1 was used to aggregate the enrichment of gene ontology (GO) terms from the following algorithms: Overrepresentation Analysis (ORA), Gene Set Enrichment Analysis (GSEA), and Gene Set Analysis (GSA). Beside the classic GO term annotation, we functionally annotated *C. elegans* neural genes using annotations from previously published reviews (*Hobert, 2013*; *Robertson and Thomas, 2006*) and used these in the same way as described for GO term analysis. These annotations were also used to extract data for particular classes of genes such as 'Synaptic release machinery' (*Figure 5A*), 'Chemoreceptors' (*Figure 5E*), and 'Neuropeptide Receptors' (*Figure 5B*). For all Supplementary files and further analysis of RNAseq data (see *Figure 5* and *Figure 5—figure supplement 1*) we applied an arbitrary cutoff of 10 transcripts per million and a *q*-value<0.05.

### Tissue enrichment analysis, heat maps, and volcano plots

Enrichment analyses were performed using the web-based software Enrichment Analysis (https://www.wormbase.org/tools/enrichment/tea/tea.cgi; *Angeles-Albores et al., 2016*). Heat maps and volcano plots showing altered gene expression in *qui-1* mutant were generated using Prism from data extracted from our custom-RNAseq analysis pipeline.

### Statistics

Statistical tests were performed using Prism. In bar graphs, error bars represent standard error of the mean (SEM). When speed plots or $Ca^{2+}$ imaging traces are shown, shaded outlines represent the SEM. Statistics of each experiment is shown in figure legends.

## Acknowledgements

We would like to thank Gemma Chandratillake and Merav Cohen for identifying mutants and José David Moñino Sánchez for his help on neurosecretion assays. We are grateful to Kaveh Ashrafi (UCSF), Piali Sengupta (Brandeis), and the *Caenorhabditis* Genetic Center (funded by National Institutes of Health Infrastructure Program P40 OD010440) for strains and reagents ...and Rebecca Butcher (Univ. Florida) for C9 pheromone. We thank Tim Stevens, Paula Freire-Pritchett, Alastair Crisp, Gurpreet

Ghattaoraya, and Fabian Amman for help with bioinformatic analysis, Ekaterina Lashmanova for help with injections, Iris Hardege for strains, and Isabel Beets (KU Leuven) and members of the de Bono Lab for comments on the manuscript. We thank the CRUK Cambridge Research Institute Genomics Core for next generation sequencing and the Flow Cytometry Facility at LMB for FACS. This research was supported by the Scientific Service Units (SSU) of IST Austria through resources provided by the Bioimaging Facility (BIF), the Life Science Facility (LSF) and Scientific Computing (SciCo-p – Bioinformatics). This work was supported by the Medical Research Council UK (Studentship to GV), an Advanced ERC grant (269,058 ACMO to MdB), and a Wellcome Investigator Award (209504/Z/17/Z to MdB).

## Additional information

### Funding

| Funder | Grant reference number | Author |
| --- | --- | --- |
| Wellcome Trust | 209504/Z/17/Z | Mario de Bono |
| H2020 European Research Council | 269058 ACMO | Mario de Bono |
| Medical Research Council | Graduate Student Fellowship | Giulio Valperga |

The funders had no role in study design, data collection, and interpretation, or the decision to submit the work for publication.

### Author contributions

Giulio Valperga, Conceptualization, Formal analysis, Investigation, Visualization, Writing – original draft, Writing – review and editing; Mario de Bono, Conceptualization, Funding acquisition, Investigation, Project administration, Supervision, Writing – original draft, Writing – review and editing

### Author ORCIDs

Giulio Valperga (iD) http://orcid.org/0000-0001-6726-3890
Mario de Bono (iD) http://orcid.org/0000-0001-8347-0443

### Decision letter and Author response

Decision letter https://doi.org/10.7554/eLife.68040.sa1
Author response https://doi.org/10.7554/eLife.68040.sa2

## Additional files

### Supplementary files

- Supplementary file 1. Genes expressed in ADL neurons.
- Supplementary file 2. Chemoreceptors expressed in ADL neurons.
- Supplementary file 3. Neuropeptide receptors expressed in ADL neurons.
- Supplementary file 4. Neuropeptides expressed in ADL neurons.
- Supplementary file 5. Genes differentially regulated in ADL neurons (Control vs *qui-1*).
- Supplementary file 6. Strain list.
- Transparent reporting form

### Data availability

Sequencing data have been deposited in GEO under accession code GSE168597.

The following dataset was generated:

| Author(s) | Year | Dataset title | Dataset URL | Database and Identifier |
| --- | --- | --- | --- | --- |
| Giulio V, Amman F, de Bono M | 2021 | Impairing one sensory modality enhances another by reprogramming peptidergic circuits in *Caenorhabditis elegans* | https://www.ncbi.nlm.nih.gov/geo/query/acc.cgi?acc=GSE168597 | NCBI Gene Expression Omnibus, GSE168597 |

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
