## [Editor Report]

In this study Valperga and de Bono make the intriguing observation that interfering with the sensory function of a nociceptive neuron, termed ADL, alters its gene expression programs causing a reconfiguration of its functions. Upon loss of its properties as a primary sensor ADL gets repurposed by oxygen sensory circuits to enhance neurosecretion in an environmental oxygen dependent manner; thereby it adopts some interneuron like properties. The study is an interesting example of cross modal plasticity in neuronal circuits. It enables future studies on the ethological function of this phenomenon.

---

## [Decision Letter]

**Decision letter after peer review:**

Thank you for submitting your article "Impairing one sensory modality enhances another by reprogramming peptidergic circuits in *Caenorhabditis elegans*" for consideration by *eLife*. Your article has been reviewed by 3 peer reviewers, one of whom is a member of our Board of Reviewing Editors, and the evaluation has been overseen by Piali Sengupta as the Senior Editor. The following individual involved in review of your submission has agreed to reveal their identity: Ilona C Grunwald Kadow (Reviewer #2).

Essential revisions:

(1) The reviewers are not convinced that the results should be interpreted as evidence for cross modal plasticity (see public review of reviewer #3). The main concern is that the qui-1 mutant background does not distinguish between a specific effect on ADL's sensory properties and a more general switch that concomitantly upregulates neurosecretion and thereby elevated behavioural responses to oxygen. If the authors insist in their interpretation, an orthogonal approach should be taken. A cell specific knockdown of ocr-2 in ADL, either via RNAi or a transgenic Cre-lox strategy, would be a more specific way to disrupt the sensory properties of ADL. Such a manipulation should lead to the same effect, i.e. up-regulation of neurosecretion and elevated behavioural responses to 21% oxygen.

(2) The authors did not exclude the possibility that other oxygen sensory neurons, e.g. AQR, PQR or BAG show altered responses in qui-1 mutants and could thereby explain the enhanced behavioral responses. In addition to URX and RMG, they should record AQR, PQR and BAG activity in wild-type and qui-1 mutants.

(3) The authors suggest a rather detailed model while providing only partial evidence for it. The authors convincingly show that neurosecretion from ADL is up-regulated in qui-1 mutants, but this could be a sheer correlation with the behavioral phenotype.

They should perform ADL cell ablations and in addition interfere with ADL neurosecretion, e.g. by expressing tetanus toxin, both in in qui-1 mutant background. The model predicts that these manipulations will suppress the enhanced behavioral responses to high oxygen.

In this vein, the behavioral responses to 21% O2 of the strains qui1;npr-22, qui;npr-22 + ADLp:npr22 rescue, ADLp:npr-22 OE, and qui-1 tkr-1 should be analyzed to further support the model.

(4) There is a large variability in the behavioral assay. For example, in Figure 1D qui-1 mutants show an elevated 21% O2 evoked response, however in Figure 1-S1B the mutants' major phenotype seems to be reduced speed at 7% O2. This would have a strong implication in how to interpret the data and requires further discussion. Related to this, quantifying locomotion speed responses as fold change with respect to the 7% condition cannot distinguish between effects on baseline speed at 7% and effects on 21% evoked response magnitude. Additional quantifications of absolute speed should be provided.

(5) The authors start their introduction by explaining how the loss of vision, for instance, can lead to the repurposing of neurons and even the rewiring of neural circuits. The presented data, however, describes a cell-autonomous mechanism that increases neuropeptide receptor expression and thereby increases peptide release. The exact relationship between (1) increased O2 sensitivity, (2) increased receptor expression and (3) increased peptide release is not fully explained by the data. This is also evidenced by their model in Figure 6. I suggest that the authors still integrate all of their findings in their model but indicate what remains not fully elucidated.

For instance, we suggest including URX etc. into your model, because this is the gained sensory modality that makes the whole paper more interesting.

(6) The issue of possible background mutations influencing the ADL transcriptomic analysis should be addressed. Were the mutants sufficiently outcrossed? This point could be addressed by providing additional transcriptome data from the strains generated in response to (1), different alleles or fully outcrossed lines.

(7) It should be stated whether the qui-1 mutant was derived from an N2 or flp-21 background, or in other words, what is the wild type.

The label wild type (WT) is applied at least to 4 different genomic strains (Control strain N2, DAF-28::mCherry, IDA-1::GFP and srh-220p:mKate). This can be confusing for the audience and it is not specified in the methods section. A table of the strains used in the study would be much appreciated. Should be also mentioned that N2 is not a 'wild' strain.

Related: The label wild type (WT) is applied at least to 4 other different genomic strains (Control strain N2, DAF-28::mCherry, IDA-1::GFP and srh-220p:mKate). This can be confusing for the audience and it is not specified in the methods section. A table of the strains used in the study would be much appreciated. Should be also mentioned that N2 is not a 'wild' strain.

(8) It would be helpful if the authors could comment about why they think that ADL O2 responses disappear in qui-1 mutants, and how O2 information received through NPR-22 directly induces enhanced neuropeptide release without involving any detectable calcium signals.

(9) In a previous research article of the lab, they showed strong responses in ADL to 21% oxygen in npr-1 background, and that ADL is unresponsive in N2 background. Contrarily, there seems to be a significant response in Figure 2E. This should be at least mentioned and discussed.

(10) we am not a fan of the work 'reprogram'. We implicates an epigenetic mechanism, and we don't think you've shown that. We suggest finding another expression.

(11) Two of the presented mutants are loss-of-functions of genes expressed in glia cells. I actually found this aspect very interesting given that these mutants phenocopy qui-1 in several aspects. We would like to hear more in the discussion about how the authors interpret these data.

(12) Figure 1: the panels don't correspond to the text (e.g., Figure 1A, B and D)

(13) The term hyperpolarization could be misleading in the context of calcium imaging (e.g. Figure 2 and associated text).

(14) Would a single copy fig-1 gDNA rescue fig-1?

(15) "These data suggest that ADL neurons release more DCVs in bbs, wrt-6, and fig-1 mutants than wild type animals in response to input from the O2 circuit." The response to the O2 circuit was directly shown only for bbs-7 (in addition to qui-1).

(16) Figure 5A: we assume each column is an independent repetition?

Also, perhaps we got this wrong, but the scales in Figure 5 and S5 show a negative log2 fold change when the expression level had actually increased from WT to qui-1?

---

## [Author Response]

Essential revisions:(1) The reviewers are not convinced that the results should be interpreted as evidence for cross modal plasticity (see public review of reviewer #3). The main concern is that the qui-1 mutant background does not distinguish between a specific effect on ADL's sensory properties and a more general switch that concomitantly upregulates neurosecretion and thereby elevated behavioural responses to oxygen. If the authors insist in their interpretation, an orthogonal approach should be taken. A cell specific knockdown of ocr-2 in ADL, either via RNAi or a transgenic Cre-lox strategy, would be a more specific way to disrupt the sensory properties of ADL. Such a manipulation should lead to the same effect, i.e. up-regulation of neurosecretion and elevated behavioural responses to 21% oxygen.

We thank our reviewers for their comments and suggestions. We have sought to test our model with a functional experiment that selectively disrupts sensory input into the ADL neurons. To achieve this, we decided to knock down a protein required for intraflagellar transport, OSM-6, rather than the OCR-2 TRP channel subunit. OCR-2 mediates not only pheromone responses in ADL, but also O_2_-escape behaviour (de Bono et al., 2002). This may reflect a broader role for OCR-2 in ADL than sensory transduction. Disrupting OSM-6 truncates sensory cilia and severely compromises many chemosensory responses, but only weakly reduces aggregation and O_2_ responses.

To target OSM-6 degradation specifically to the ADL neurons we knocked in DNA encoding an Auxin Inducible Degron (AID) into the *osm-6* locus, and expressed TIR1 in ADL to achieve cell-specificity. TIR1 is required for AID. We have added the new data to Figure 4F–G and Figure 4—figure supplement 2. We show that expressing TIR1 in ADL disrupts OSM-6::AID function both in the presence and absence of Auxin. This agrees with recent work that tested the efficiency and specificity of the AID system (Hills-Muckey et al., 2021). A partial OSM-6::AID reduction in ADL recapitulates many of the phenotypes of *qui-1* mutants, including increased neurosecretion from ADL, heightened ADL responses to O_2_ inputs and a small but significant enhancement of the O_2_-escape response. We think these new data support our interpretation that a change in ADL’s sensory properties leads to heightened response of ADL neurons to O_2_ inputs, a phenotype observed in *qui-1* and multiple other sensory defective mutants and a hallmark of cross-modal plasticity. However, the effects of knocking down *osm-6* on ADL function also appear to be complex, as the stronger *osm-6* knockdown achieved by adding auxin to the *osm-6::AID* knockin animals expressing TIR1 in ADL, unexpectedly gives weaker phenotypes than when auxin is absent.

(2) The authors did not exclude the possibility that other oxygen sensory neurons, e.g. AQR, PQR or BAG show altered responses in qui-1 mutants and could thereby explain the enhanced behavioral responses. In addition to URX and RMG, they should record AQR, PQR and BAG activity in wild-type and qui-1 mutants.

Our reviewers are right that we did not probe the contribution of other O_2_ sensors such as AQR, PQR and BAG. We focused on URX and RMG since they are connected with ASH and ADL neurons in a hub-and-spoke circuit. We have now imaged O_2_-evoked Ca^2+^ responses in AQR and PQR in *qui-1* mutants using the Ca^2+^ sensor YC3.60, and incorporated the data in Figure 1—figure supplement 1D and 1E. In both neurons, the responses of *qui-1* and wild-type animals are comparable. This suggests the enhanced O_2_-escape behaviour of *qui-1* mutants is not explained by increased activity of URX, AQR, PQR or RMG O_2_-sensing neurons.

We regret we have not been able to image BAG’s O_2_-evoked Ca^2+^ responses. These responses are smaller than in other O_2_ sensors and require a different microscope set-up capable of higher magnification, and we have encountered technical difficulties in setting up this system.

(3) The authors suggest a rather detailed model while providing only partial evidence for it. The authors convincingly show that neurosecretion from ADL is up-regulated in qui-1 mutants, but this could be a sheer correlation with the behavioral phenotype.They should perform ADL cell ablations and in addition interfere with ADL neurosecretion, e.g. by expressing tetanus toxin, both in in qui-1 mutant background. The model predicts that these manipulations will suppress the enhanced behavioral responses to high oxygen.In this vein, the behavioral responses to 21% O2 of the strains qui1;npr-22, qui;npr-22 + ADLp:npr22 rescue, ADLp:npr-22 OE, and qui-1 tkr-1 should be analyzed to further support the model.

Our reviewers are correct that in our model we speculated as to why *qui-1* mutants display enhanced O_2_-escape behaviour. We have now updated the Discussion to clarify that this mechanism remains unclear, especially in the light of further experiments suggested by our reviewers.

Our work identified multiple mutants that concomitantly exhibit O_2_-evoked DCV release from ADL neurons and enhanced O_2_-escape behaviour (*qui-1, bbs-7*, *wrt-6* and *fig-1* mutants). Follow up experiments with *osm-6::AID* knockin animals that selectively perturb ADL cilia function by expressing TIR1 in this neuron recapitulate these phenotypes (See Point 1). As suggested by our reviewers, we probed the molecular mechanism linking ADL output to O_2_-escape behaviour. We disrupted ADL neurosecretion in *qui-1* mutants by selectively expressing tetanus toxin (TeTx) in this neuron, and assayed O_2_-evoked escape behaviour (Figure 5—figure supplement 2D). The TeTx expressing *qui-1* animals retained elevated O_2_-escape; this suggests that elevated neurosecretion does not, on its own, explain their altered behaviour. We previously showed that ablating ADL and ASH neurons disrupts aggregation in *npr-1* animals (de Bono et al., 2002), confirming that these neurons are important for this behaviour. We had also shown that expressing TeTx in ASH and ADL neurons in *npr-1* mutants has only a small effect on O_2_-escape response, although it reduced by about half the reversal response evoked by 3 mM Cu^2+^ (Laurent et al., 2015).

We also did not observe any change in O_2_-escape behaviour in *qui-1;npr-22* or *qui-1;tkr-1* double mutants compared to *qui-1* controls, or in transgenic animals over-expressing NPR-22 compared to wild-type (Figure 5—figure supplement 2A-C). Together, these data suggest removing *npr-22*, *tkr-1* or blocking ADL’s synaptic output, is not sufficient to account for the O_2_-responses of *qui-1* mutants. Most likely, communication through gap junctions is also important for the O_2_-escape response of *qui-1* mutants. Further experiments are clearly required to test this model.

(4) There is a large variability in the behavioral assay. For example, in Figure 1D qui-1 mutants show an elevated 21% O2 evoked response, however in Figure 1-S1B the mutants' major phenotype seems to be reduced speed at 7% O2. This would have a strong implication in how to interpret the data and requires further discussion. Related to this, quantifying locomotion speed responses as fold change with respect to the 7% condition cannot distinguish between effects on baseline speed at 7% and effects on 21% evoked response magnitude. Additional quantifications of absolute speed should be provided.

Our reviewers are right in pointing out that there is some variability in animals’ speed during O_2_-escape assays. This is something we have observed before. The biggest source of variability appears to be the thickness of the bacterial lawn on which animals are either grown or assayed. For this reason, all our assays are day-matched – different genotypes are grown together and assayed on the same day, on plates seeded at the same time. All our Figures also include data obtained on at least three different days.

The reviewers also correctly point out that *qui-1* mutants and N2 animals differ in their responses to both 21% and 7% O_2_: not only are *qui-1* animals more strongly aroused by 21% O_2_ but they also move more slowly than WT at 7% O_2_. Other aggregation mutants, including *npr-1*, show these same phenotypic differences from N2. Disrupting the O_2_ responses of *npr-1* animals, for example by deleting genes encoding the molecular O_2_ sensors GCY-35 and GCY-36, confers N2-like behavior: *npr-1;gcy-35* and *npr-1*;*gcy-36* animals move slower than *npr-1* at 21% O_2_ but faster at 7% O_2_. We think the intermediate speed of N2 animals (and *gcy-35;npr-1* mutants etc) at 7% O_2_ reflects circuit plasticity; the persistent low activity of the O_2_-sensing circuit in these animals renders the locomotory circuit more sensitive to input from other circuits. By contrast, in *npr-1* and *qui-1* mutants kept at 21% O_2_ the locomotory circuit is adapted to persistent high input form the O_2_ sensing neurons; when this input is removed, by shifting animals to 7% O_2_, the locomotory circuit is less sensitive to input from other circuits – leading to animals moving slowly. Consistent with this interpretation, *npr-1* animals kept for long periods at 7% O_2_ gradually begin to move faster at this O_2_ concentration. Moreover, when such animals are stimulated by 21% O_2_, they reach significantly higher speeds than animals habituated to 21% O_2_.

This is why we think the ratio of *C. elegans*’ speed at 21% O_2_ and 7% O_2_ is a good proxy for the functionality of the O_2_ circuit in regulating the locomotory circuit.

We acknowledge that the circuits controlling locomotory activity are complex, and regulated by sensory modalities other than O_2_. However, we think the observations described above justify using the change in the animals’ speed as we switch them from low to high O_2_ as a proxy for changes in the activity of the O_2_ circuit. We have outlined this reasoning in the Method Section.

(5) The authors start their introduction by explaining how the loss of vision, for instance, can lead to the repurposing of neurons and even the rewiring of neural circuits. The presented data, however, describes a cell-autonomous mechanism that increases neuropeptide receptor expression and thereby increases peptide release. The exact relationship between (1) increased O2 sensitivity, (2) increased receptor expression and (3) increased peptide release is not fully explained by the data. This is also evidenced by their model in Figure 6. I suggest that the authors still integrate all of their findings in their model but indicate what remains not fully elucidated.For instance, we suggest including URX etc. into your model, because this is the gained sensory modality that makes the whole paper more interesting.

We thank the reviewers for their comment. We agree that our model suggests that cell-autonomous changes in ADL alters ADL responsiveness to input from other neurons, notably URX. We have updated our model in Figure 6 to indicate O_2_-sensory neurons (including URX). We have also added question marks to our model to highlight unknown molecular mechanisms that need to be investigated in future work. Namely, the molecular mechanisms that reconfigures the expression profile of ADL, and the signal from O_2_-sensing neurons that supports ADL’s O_2_-evoked neurosecretion after sensory impairment.

(6) The issue of possible background mutations influencing the ADL transcriptomic analysis should be addressed. Were the mutants sufficiently outcrossed? This point could be addressed by providing additional transcriptome data from the strains generated in response to (1), different alleles or fully outcrossed lines.

We worried about unspecific effects of background mutations both on the ADL transcriptome and on other *qui-1* related phenotypes. We regret we did not explicitly address this point in our initial submission. To remove background mutations, mutants isolated in our screen, including *qui-1*, were backcrossed with the N2 laboratory strain a minimum of four times. These *qui-1* animals were further crossed into a 5 times outcrossed line that expresses the fluorescent protein mKate specifically in ADL, to generate the strains from which we sorted ADL neurons by FACS. Mutant and transgenic strains were outcrossed using the N2 laboratory strain. We explain this in the Methods section of the revised manuscript.

The extensive outcrossing make us confident that the large majority of differentially regulated genes between wild-type and *qui-1* samples in ADL are due to the absence of *qui-1*. Supporting this, both mutations in neuropeptide receptors identified by our profiling, *npr-22* and *tkr-1*, suppress ADL’s elevated neurosecretion. Nevertheless, we have added a note to explicitly bring up the concern raised by our reviewers, that some transcriptional differences could be the result of background mutations.

(7) It should be stated whether the qui-1 mutant was derived from an N2 or flp-21 background, or in other words, what is the wild type.The label wild type (WT) is applied at least to 4 different genomic strains (Control strain N2, DAF-28::mCherry, IDA-1::GFP and srh-220p:mKate). This can be confusing for the audience and it is not specified in the methods section. A table of the strains used in the study would be much appreciated. Should be also mentioned that N2 is not a 'wild' strain.Related: The label wild type (WT) is applied at least to 4 other different genomic strains (Control strain N2, DAF-28::mCherry, IDA-1::GFP and srh-220p:mKate). This can be confusing for the audience and it is not specified in the methods section. A table of the strains used in the study would be much appreciated. Should be also mentioned that N2 is not a 'wild' strain.

We isolated the *qui-1* mutant from the N2 parental strain. We have now inserted a line in the Results section of the revised manuscript to say this.

We agree that our use of “WT” is misleading. We used this term in an attempt to keep figures accessible since *C. elegans* genetic nomenclature can be confusing to non-specialist readers. To address this issue and ensure we do not mislead our readers, we opted for the term “Control” in Text and Figures while referring to strains we used to image Ca^2+^ responses, ADL neurosecretion, IDA-1 localisation and analyses of ADL profiling, while also reporting the genotype of each control line in the respective Figure legends. We have also compiled a Strain List (Supplementary File 6) listing the genotype of all strains used in the paper and detailing the figure in which they have been used, to help readers track different control strains.

We also highlight in the revised version of the Results section that the N2 laboratory strain has accumulated mutations since the original wild strain was domesticated more than 60 years ago.

(8) It would be helpful if the authors could comment about why they think that ADL O2 responses disappear in qui-1 mutants, and how O2 information received through NPR-22 directly induces enhanced neuropeptide release without involving any detectable calcium signals.

We can only speculate why O_2_-evoked responses in ADL disappear in *qui-1* mutants. One possibility is that ADL becomes less excitable due to the reconfigured gene expression associated with loss of *qui-1* in ADL. This model would predict that selectively knocking down *qui-1* in ADL would confer the same Ca^2+^ response phenotype. Blocking ADL neurosecretion with TeTx in *qui-1* mutants would test if the increased ADL neurosecretion we describe feeds back to reduce the O_2_-evoked Ca^2+^ response in ADL. An alternative hypothesis is that the effect of disrupting *qui-1* is non-cell-autonomous, altering excitatory or inhibitory input to ADL from other *qui-1* expressing neurons. We have not tested if neurosecretion from other *qui-1*-expressing neurons is altered in *qui-1* mutants.

Strikingly, while disrupting *qui-1* leads to loss of a measurable O_2_-evoked Ca^2+^ response in ADL, these neurons display elevated O_2_-evoked neurosecretion in *qui-1* mutants. This implies that some O_2_-evoked Ca^2+^ responses are retained in ADL’s axons in *qui-1* mutants. It also suggests that other second messengers upregulate neurosecretion. Elevating cAMP, for example, can promote dense-core vesicle release more efficiently than increasing Ca^2+^ levels (Steuer Costa et al., 2017). Altered G-protein coupled receptor signalling could lead to elevated cAMP levels and increased neurosecretion in *qui-1* mutants. It is worth noting that in N2 controls, ADL does not display O_2_-evoked neurosecretion despite showing measurable Ca^2+^ responses.

(9) In a previous research article of the lab, they showed strong responses in ADL to 21% oxygen in npr-1 background, and that ADL is unresponsive in N2 background. Contrarily, there seems to be a significant response in Figure 2E. This should be at least mentioned and discussed.

Our reviewers are correct to note the difference. This likely reflects use of improved Ca^2+^ sensors. Fenk *et al.* imaged ADL’s O_2_-evoked Ca^2+^ response in the N2 laboratory strain and *npr-1* mutants using GCaMP3; in this paper we use GCaMP6s an improved version of GCaMP3 (Tian et al., 2009). The 10-fold increase in signal intensity given by GCaMP6s compared to GCaMP3 likely explains why our imaging experiments detect Ca^2+^ responses not observed by Fenk *et al.* We have added a note to say this.

(10) we am not a fan of the work 'reprogram'. We implicates an epigenetic mechanism, and we don't think you've shown that. We suggest finding another expression.

We borrowed the term “reprogramming” to denote a change in ADL’s proprieties, but did not wish to imply that epigenetic mechanisms were involved. Given that what we report changes in gene expression, we can see that such a misunderstanding could easily arise. To avoid confusion, we have removed the word “reprogramming” from the revised manuscript, and simply refer to changes in gene expression.

(11) Two of the presented mutants are loss-of-functions of genes expressed in glia cells. I actually found this aspect very interesting given that these mutants phenocopy qui-1 in several aspects. We would like to hear more in the discussion about how the authors interpret these data.

We agree that it is interesting that mutants for two genes expressed in glial cells, *wrt-6* and *fig-1*, phenocopy *qui-1*. We have revised the Discussion to expand on this observation.

(12) Figure 1: the panels don't correspond to the text (e.g., Figure 1A, B and D)

We apologise for this oversight. We have corrected the text.

(13) The term hyperpolarization could be misleading in the context of calcium imaging (e.g. Figure 2 and associated text).

We have corrected the text. We now state that while in wild-type animals O_2_ stimulation increases ADL’s Ca^2+^ levels, in *qui-1* mutants the same stimulus seems to reduce Ca^2+^ levels.

(14) Would a single copy fig-1 gDNA rescue fig-1?

Yes, our prediction would be that a single copy insertion of the wild-type *fig-1* gene would rescue phenotypes associated with this mutant. In the absence of this rescue experiment, our analysis of multiple *fig-1* alleles gives us confidence that impairing *fig-1* increases O_2_-escape behaviour.

(15) "These data suggest that ADL neurons release more DCVs in bbs, wrt-6, and fig-1 mutants than wild type animals in response to input from the O2 circuit." The response to the O2 circuit was directly shown only for bbs-7 (in addition to qui-1).

Thank you for pointing out this misleading statement. We have reworded this statement to say that *bbs-7* and *qui-1* mutants increased ADL neurosecretion reflects increased responsiveness to O_2_ input. We also suggest that enhanced neurosecretion from ADL in other sensory defective mutants characterised in this study may similarly reflect increased responsiveness to O_2_-input. We believe data from our OSM-6::AID knockdown experiments support this interpretation.

(16) Figure 5A: we assume each column is an independent repetition?Also, perhaps we got this wrong, but the scales in Figure 5 and S5 show a negative log2 fold change when the expression level had actually increased from WT to qui-1?

That is correct. We apologise for the oversight. Figure legends now state that each column in heat maps represents a biological replicate.

References:

de Bono M, Tobin DM, Davis MW, Avery L, Bargmann CI. 2002. Social feeding in *Caenorhabditis elegans* is induced by neurons that detect aversive stimuli. Nature 419:899–903. doi:10.1038/nature01169

Steuer Costa W, Yu S, Liewald JF, Gottschalk A. 2017. Fast cAMP Modulation of Neurotransmission via Neuropeptide Signals and Vesicle Loading. Curr Biol 27:495–507. doi:10.1016/j.cub.2016.12.055

Hills-Muckey K, Martinez MAQ, Stec N, Hebbar S, Saldanha J, Medwig-Kinney TN, Moore FEQ, Ivanova M, Morao A, Ward JD, Moss EG, Ercan S, Zinovyeva AY, Matus DQ, Hammell CM. 2021. An engineered, orthogonal auxin analog/AtTIR1(F79G) pairing improves both specificity and efficacy of the auxin degradation system in *Caenorhabditis elegans*. Biorxiv 2021.08.06.455414. doi:10.1101/2021.08.06.455414

Laurent P, Soltesz Z, Nelson GM, Chen C, Arellano-Carbajal F, Levy E, de Bono M. 2015. Decoding a neural circuit controlling global animal state in *C. elegans*. *eLife* 4:e1004156. doi:10.7554/*eLife*.04241

Tian L, Hires SA, Mao T, Huber D, Chiappe ME, Chalasani SH, Petreanu L, Akerboom J, McKinney SA, Schreiter ER, Bargmann CI, Jayaraman V, Svoboda K, Looger LL. 2009. Imaging neural activity in worms, flies and mice with improved GCaMP calcium indicators. Nature Methods 6:875–881. doi:10.1038/nmeth.1398